# Optimal Fair Aggregation of Crowdsourced Noisy Labels using Demographic Parity Constraints

**Gabriel Singer** [* 1]   **Samuel Gruffaz** [* 1]   **Olivier Vo Van** [2]   **Nicolas Vayatis** [1]   **Argyris Kalogeratos** [1]

## Abstract

As acquiring reliable ground-truth labels is usually costly, or infeasible, crowdsourcing and aggregation of noisy human annotations is a common alternative. Aggregating subjective labels, though, may amplify individual biases, particularly regarding sensitive features, raising fairness concerns. Nonetheless, fairness in crowdsourced aggregation remains largely unexplored, with no existing convergence guarantees and only limited post-processing approaches for enforcing $\varepsilon$-fairness under demographic parity. We address this gap by analyzing the fairness of crowdsourced aggregation methods within the $\varepsilon$-fairness framework, for Majority Vote and Optimal Bayesian aggregation. In the small-crowd regime, we derive an upper bound on the fairness gap of Majority Vote in terms of the fairness gaps of the individual annotators. We further show that the fairness gap of the aggregated consensus converges exponentially fast to that of the ground-truth under interpretable conditions. Since ground-truth itself may still be unfair, we generalize a state-of-the-art multiclass fairness post-processing algorithm from the continuous to the discrete setting, which enforces strict demographic parity constraints on any aggregation rule. Experiments on synthetic and real datasets demonstrate the effectiveness of our approach and corroborate the theoretical insights.

## 1. Introduction

Crowdsourcing (Ibrahim et al., 2025) has become a standard paradigm for constructing large-scale datasets in domains where ground-truth is inherently subjective or prohibitively expensive to obtain, such as medical diagnosis, content moderation, and sentiment analysis. There, each item is typically labeled by multiple annotators , resulting in a collection of noisy labels rather than a single reliable annotation.

Two main paradigms have been developed to leverage crowdsourced labels. (i) Label integration, also known as label aggregation or truth discovery, aims to infer the latent ground-truth label by aggregating multiple annotations while explicitly modeling annotator reliability. (ii) End-to-end learning, where noisy labels are directly used to train a classifier through loss functions or architectures that explicitly account for label noise. While both paradigms have been extensively studied (Guo et al., 2023; 2024; Nguyen et al., 2024), recent works have highlighted that issues of bias and fairness remain largely under-investigated in the crowdsourcing literature (Ibrahim et al., 2025; Lazier et al., 2023, p.19). In practice, annotators often exhibit systematic biases correlated with sensitive features such as gender or race. When label errors are structured rather than random, the classical statistical benefits of aggregation become unclear: although increasing the crowd size reduces variance, its impact on fairness is not well-understood.

In particular, Lazier et al. (2023) demonstrate through extensive experiments that enforcing fairness solely at the classifier level, after aggregating labels in a fairness-agnostic manner, is insufficient. Instead, there is a need for label integration methods that explicitly respect fairness constraints with regard to sensitive features. To the best of our knowledge, only the work of Li et al. (2020) directly addresses fairness in the crowdsourcing setting using a demographic a parity constraint, and it is shown to be state-of-the-art in Lazier et al. (2023). However, this approach lacks theoretical guarantees: the proposed algorithm is not proven to converge, and some modeling choices are questionable; for instance, assuming Gaussian noise despite the discrete nature of labels.

In the present work, we aim to bridge both the theoretical and practical gaps in fair crowdsourced aggregation. We focus on *demographic parity* as our fairness criterion, which, despite being simple, remains the most widely-used fairness notion in practice (Hort et al., 2024). It is also relatively

---

[*]Equal contribution  [1]University Paris Saclay, ENS Paris Saclay, Centre Borelli [2]SNCF, Paris, France. Correspondence to: Gabriel Singer <gabriel.singer@ens-paris-saclay.fr>, Samuel Gruffaz <samuel.gruffaz@ens-paris-saclay.fr>.

*Proceedings of the 43rd International Conference on Machine Learning*, Seoul, South Korea. PMLR 306, 2026. Copyright 2026 by the author(s).

robust to contextual bias (Fawkes et al., 2024) and, crucially, does not require access to ground-truth labels for estimation, making it particularly suited to the crowdsourcing setting.

**Contributions.** Our theoretical and methodological contributions are summarized as follows:

- *Theoretical Analysis:* By simplifying the conditions of the convergence results of Gao et al. (2016), we derive the first sharp, non-asymptotic bounds on the fairness gap for both Majority Vote and Bayesian Aggregation, together with an intuitive interpretation (Theorem 3.2). In particular, we show that both methods converge exponentially fast to the fairness gap of the ground-truth label, provided that the errors of low-quality annotators are compensated by sufficiently reliable ones. In the small-crowd regime, we further upper-bound the fairness gap of Majority Vote (Theorem 3.6) in terms of the individual fairness gaps of the annotators, using optimal inequalities for Poisson binomial distributions (Baillon et al., 2015).
- *Framework Adaptation:* Building on the $\varepsilon$-fairness framework of Denis et al. (2024) for post-processing classifiers under demographic parity constraints, we generalize its main results to the crowdsourcing setting for binary labels (Theorem 4.1) and multi-class labels (Theorem A.1). In particular, we extend the theory to discrete inputs, whereas the original framework was restricted to continuous ones (Section 4).
- *Algorithm:* To overcome the theoretical and practical limitations identified above, we introduce *FairCrowd*, a post-processing algorithm that regularizes any label aggregation rule to enforce strict $\varepsilon$-fairness constraints. Our method achieves state-of-the-art performance (Section 5) on synthetic data, as well as on two benchmark crowdsourcing datasets: *Crowd Judgement* and *Jigsaw Toxicity*.

**Conflict of Interest Disclosure.** The authors declare that they have no conflicts of interest relevant to the content of this paper.

## 1.1. Related work

**Crowdsourcing.** While Majority Vote and Weighted Majority Vote are among the simplest aggregation rules, the *Bayes-optimal classifier/aggregator* (Nitzan & Paroush, 1982; Berend & Kontorovich, 2014; Gao et al., 2016) is optimal with respect to the 0-1 risk. However, estimating the *Bayes-optimal aggregation* requires access to gold-standard labels to estimate annotator confusion probabilities, which are often unavailable in practice. This limitation has motivated the development of alternative computable aggregation methods, including approaches based on the Expectation-Maximization (EM) algorithm (Raykar et al., 2010), or non-negative matrix factorization (Ibrahim et al., 2025). Estimating annotator confusion probabilities without ground-truth labels raises identifiability issues, particularly

when these probabilities depend on task features.

While most crowdsourcing methods are evaluated primarily in terms of accuracy, their fairness properties have only recently been studied (Lazier et al., 2023). Beyond aggregation, fairness in crowdsourcing systems has also been examined in the context of hiring problems, where the goal is to select annotators in a fair manner while maintaining high accuracy under budget constraints (Goel & Faltings, 2019; Li et al., 2020).

**Fairness.** Recent empirical studies have shown that human annotators often project societal stereotypes onto their labels, leading to biases correlated with sensitive features such as race or gender (Sap et al., 2019). Moreover, as highlighted by Geva et al. (2019), annotators' individual identities and backgrounds can substantially influence the resulting data distributions, indicating that crowdsourced labels are not objective and rather reflect the collective subjective biases of the crowd. Optimal $\varepsilon$-fair classifiers under parity constraints have been studied in (Denis et al., 2024; Xian et al., 2023; Zeng et al., 2022). While Denis et al. (2024) adopt a min-max formulation, Xian et al. (2023) rely on optimal transport; both approaches provide $\varepsilon$-fair classifiers in the multiclass setting, whereas (Zeng et al., 2022) is restricted to the binary case. Compared to the work of Xian et al. (2023), Denis et al. (2024) offer a lighter statistical analysis and a more computationally efficient implementation.

## 2. Problem Setting

For simplicity, we consider a binary label $Y \in \{0, 1\}$, but we clarify when results can be generalized. Instead of the ground-truth label $Y$, we have access to a set of $R$ noisy labels $\tilde{Y}_{1:R} = (\tilde{Y}_1, ..., \tilde{Y}_R) \in \{0, 1\}^R$, provided by a crowd of annotators. The annotators are influenced by context, namely task features $X \in \bar{\mathcal{X}}$, and may often be biased according to context. We focus on the special case where this is a binary sensitive feature $A \in \{0, 1\}$ within the features, such that $\bar{\mathcal{X}} = \mathcal{X} \times \{0, 1\}$ where $\mathcal{X}$ is a set of non-sensitive features. The sensitive feature $A$ might be for instance gender, race or weather conditions, e.g. raining or not, in sensor industrial context. It corresponds to standard fairness settings, therefore constitutes a natural first step toward understanding aggregation in the presence of contextual features. Note that $\tilde{Y}_{1:R}, X, A, Y$ are random variables defined on the same probability space $(\Omega, \mathcal{A}(\Omega), \mathbb{P})$, but taking different values.

The goal is to design an aggregation rule:

$$\phi : \{0, 1\}^R \times \mathcal{X} \times \{0, 1\} \mapsto \{0, 1\},$$

which maps the noisy labels, the features, and the sensitive feature to a single predicted label $\hat{Y}^\phi := \phi(\tilde{Y}_{1:R}, X, A)$.

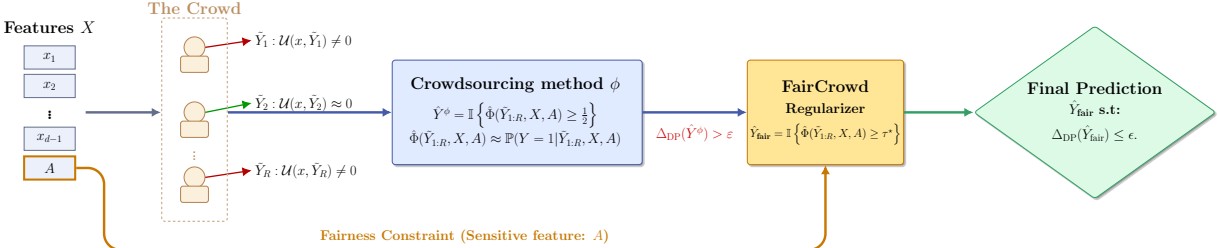

*Figure 1.* Overview of our FairCrowd algorithm.

In the crowdsourcing setting, the ground-truth $Y$ is latent and generally unavailable (Whitehill et al., 2009; Welinder et al., 2010). We adopt *Demographic Parity* (DP) as our fairness criterion, as it does not require access to ground-truth labels and is therefore natural in our setting.

Since perfect fairness is rarely achievable in practice, we choose the relaxed notion of $\varepsilon$*-fairness*.

**Definition 2.1** (Global DP Gap & $\varepsilon$-Fairness). The *Global Demographic Parity* gap is defined as:

$$\Delta_{\mathrm{DP}}(\tilde{Y}) := \big|\mathbb{P}(\tilde{Y}=1\,|\,A=1)-\mathbb{P}(\tilde{Y}=1\,|\,A=0)\big|. \quad (1)$$

A predictor $\tilde{Y}$ is said to be $\varepsilon$*-fair* if $\Delta_{\mathrm{DP}}(\tilde{Y}) \leq \varepsilon$.

**The global DP is counter-intuitive.** Let us define $Q_a(x) := \mathbb{P}(\tilde{Y}=1\,|\,X=x, A=a)$. Then, $\mathcal{U}(\tilde{Y}, x) := |Q_1(x) - Q_0(x)|$ is the *local demographic disparity*, for any $(x,a) \in \mathcal{X} \times \{0,1\}$ and binary random variable $\tilde{Y}$. Even if an annotator is perfectly pointwise fair, i.e. $\mathcal{U}(\tilde{Y}, x) = 0$ for any $x \in \mathcal{X}$, the demographic parity can still be higher than expected when the feature distributions differ between groups, as detailed below.

**Lemma 2.2.** *Let $\tilde{Y}$ be a binary random variable such that the annotator is perfectly pointwise fair, i.e. $\mathcal{U}(\tilde{Y}, x) = 0$ for all $x \in \mathcal{X}$. Then, there exists $\tilde{Y}$: $\Delta_{\mathrm{DP}}(\tilde{Y}) \neq 0$.*

The proof is constructive, based on Bayes' theorem; see Appendix C for more details.

## 3. Theoretical Analysis

Our analysis takes as its starting point the error probability bounds established by Gao et al. (2016), who demonstrated that under specific regularity conditions, the classification error of crowdsourced aggregation decays exponentially with the number of annotators $R$. This analysis is extended to the fairness properties of the resulting consensus by employing assumptions specifically tailored to the present context. Two aggregators are investigated: the Bayesian aggregator and the Majority Vote. For the sake of clarity, we focus our theoretical analysis on the binary classification setting. However, the proposed framework extends naturally to the multi-class regime. In what follows, we assume the *one-coin model*: for any $(a,y,x) \in \{0,1\}^2 \times \mathcal{X}$, let $p_r(a,x)$

denote the annotator skills:

$$p_r(a,x) := \mathbb{P}(\tilde{Y}_r = y\,|\,Y=y, X=x, A=a). \quad (2)$$

***Bayesian Vote:*** The aggregation function for the Bayesian Vote, $\phi : \mathcal{X} \times \{0,1\} \times \{0,1\}^R \to \{0,1\}$ is the one that minimizes the Bayes risk:

$$\mathcal{R}(\phi) := \mathbb{P}\big(\phi(\tilde{Y}_{1:R}, X, A) \neq Y\big). \quad (3)$$

The solution to this minimization problem is the *Bayes Optimal Classifier*, denoted $\phi^\star$, which predicts the label with the highest posterior probability given the observed votes and features:

$$\phi^\star(\tilde{Y}_{1:R}, X, A) := \mathbb{I}\Big\{\mathbb{P}(Y=1\,|\,\tilde{Y}_{1:R}, X, A) \geq 1/2\Big\}. \quad (4)$$

***Majority Vote:*** The most common and simple aggregator is the Majority Vote ($\phi^{\mathrm{MV}}$), which treats all annotators as equal:

$$\phi^{\mathrm{MV}}(\tilde{Y}_{1:R}, X, A) := \mathbb{I}\left\{\sum_{r=1}^{R}\tilde{Y}_r \geq \frac{R}{2}\right\}. \quad (5)$$

**Theorem 3.1.** *[(Gao et al., 2016)] For any $(x,a)$ with $R$ annotators having skills $\{p_r(x,a)\}_{r=1}^{R}$, the conditional error probability of $\phi \in \{\phi^{\mathrm{MV}}, \phi^\star\}$ satisfies:*

$$\mathbb{P}\big(\hat{Y}_R^\phi \neq Y\,|\,X=x, A=a\big) \leq \exp\big(-RK_\phi(x,a)\big),$$

*where $K_\phi(\cdot, \cdot)$ is the error exponent corresponding to the aggregator:*

• *Bayesian Vote ($\phi^\star$): The error exponent is given by:*

$$K_{\phi^\star}(a,x) = -\frac{1}{R}\sum_{i=1}^{R}\ln\Big(2\sqrt{p_i(x,a)(1-p_i(x,a))}\Big).$$

• *Majority Vote ($\phi^{\mathrm{MV}}$): The error exponent is*

$$K_{\phi^{\mathrm{MV}}}(a,x) = -\min_{t\in(0,1]}\frac{1}{R}\sum_{i=1}^{R}\ln p_i(x,a)t + \frac{1-p_i(x,a)}{t}.$$

$$(6)$$

*In the original version of the theorem by Gao et al. (2016), additional assumptions are introduced to derive a lower bound. In our setting, we retain only the minimal set of assumptions necessary, as we are exclusively interested in the upper bound.*

**Proposition 3.2.** *Let $R \in \mathbb{N}^{\star}$ denote the crowd size and $\phi \in \{\phi^{\star}, \phi^{\mathrm{MV}}\}$ be the aggregation rule. The demographic parity gap of the aggregated label $\hat{Y}_R^{\phi}$ converges to that of the ground-truth $Y$ with the non-asymptotic bound:*

$$\left| \Delta_{\mathrm{DP}}(\hat{Y}_R^{\phi}) - \Delta_{\mathrm{DP}}(Y) \right| \leq \sum_{a \in \{0,1\}} \mathbb{E}_{X \mid A = a} \left[ e^{-R \cdot K_{\phi}(a, X)} \right],$$
(7)

*where $K_{\phi}$ is defined in Theorem 3.1.*

We propose assumptions that are better adapted to our fairness setting, such that $\lim_{R \to \infty} \mathbb{E}_{X \mid A = a} \left[ e^{-R \cdot K_{\phi}(a, X)} \right]$ exists and is equal to $0$. Given that the current conditions in the literature are not easily interpretable, we propose a novel solution: divide the crowd in three disjoint groups according to their skills

$$\mathcal{G}_R = \{r : p_r(A, X) \geq 1/2 + \varepsilon \ \text{a.s.}\}, \ |\mathcal{G}_R| = g_{1,R},$$
$$\mathcal{B}_R = \{r : C \leq p_r(A, X) \leq 1/2 - \varepsilon' \ \text{a.s.}\}, \ |\mathcal{B}_R| = g_{2,R},$$
$$\mathcal{M}_R = (\mathcal{G}_R \cup \mathcal{B}_R)^c, \quad |\mathcal{M}_R| = R - g_{1,R} - g_{2,R},$$

where $(\varepsilon, \varepsilon') \in (0, 1/2)^2$. The set $\mathcal{G}_R$ consists of the competent annotators defined by the threshold $1/2 + \varepsilon$. In contrast, $\mathcal{B}_R$ contains adversarial annotators defined by the threshold $1/2 - \varepsilon'$. As the margins $\varepsilon, \varepsilon'$ approach zero, $\mathcal{G}_R$ and $\mathcal{B}_R$ might include annotators who are only marginally better or worse than random guessing. The parameter $C$ represents the lower bound on annotator accuracy; letting $C \to 0$ allows the presence of fully adversarial annotators who systematically predict the opposite of the ground-truth. Set $\mathcal{E}_R(\phi) := \limsup_{R \to \infty} \exp\left(-R \cdot K_{\phi}(A, X)\right)$.

**Lemma 3.3.** $\mathcal{E}_R(\phi^{\mathrm{MV}}) = 0$ *if:*

$$\liminf_{R \to \infty} \left( \frac{g_{1,R}}{R}(1 + 2\varepsilon) + \frac{2C g_{2,R}}{R} \right) > 1 \quad \text{a.s.} \quad (8)$$

$\mathcal{E}_R(\phi^{\star}) = 0$ *if:*

$$\sum_{r=1}^{\infty} \left( p_r(X, A) - \frac{1}{2} \right)^2 = \infty \quad \text{a.s.} \quad (9)$$

**Interpretations.** To give more insights to the condition (8) consider the worst-case scenario, where non-expert annotators are allowed to be adversarial (i.e. $C = 0$). Then, the condition simplifies to $\frac{g_{1,R}}{R} > \frac{1}{1+2\varepsilon}$. As the margin of the experts decreases ($\varepsilon \to 0$), the required proportion $\frac{g_{1,R}}{R}$ approaches 1. This implies that to compensate for "weak" experts in an adversarial environment, the crowd must be

composed almost entirely of experts to guarantee convergence. Concerning the Bayesian Vote, condition (9) says that there has to be a positive probability that an annotator has skills that are not purely random. This is actually a lighter condition than the one for Majority Vote.

**Theorem 3.4** (Asymptotic Fairness Consistency)**.** *Let $\phi \in \{\phi^{\star}, \phi^{\mathrm{MV}}\}$. Under Condition (8) for Majority Vote, and (9) for the Bayesian Vote, the aggregated consensus is asymptotically $Y$-fair:*

$$\lim_{R \to \infty} \Delta_{\mathrm{DP}}(\hat{Y}_R^{\phi}) = \Delta_{\mathrm{DP}}(Y).$$

The proof relies on the Dominated Convergence Theorem. The primary technical challenge lies in deriving the interpretable sufficient conditions (8) and (9). Establishing these conditions requires a sharp lower bound on the error exponent $K_{\phi}$, the analysis of which relies on the equicontinuity of the associated sequence of functions.

$\epsilon$**-fairness of Majority Vote.** Next, we provide an explicit $\varepsilon$ fairness bound for the Majority Vote as a function of the number of annotators $R$. Define for any given $r \in [R]$ and Bernoulli random variable $Y_r$; $l_r := \mathbb{P}(Y_r = 1)$.

**Lemma 3.5.** *Let $Y_1, ..., Y_R$ be independent Bernoulli random variables with parameters $l_1, ..., l_R$. Define*

$F : (l_1, ..., l_R) \mapsto \mathbb{P}\left( \sum_{r=1}^{R} Y_r > R/2 \right)$. *Then, for every $r \in [R]$, we have:*

$$\frac{\partial F}{\partial l_r} = \mathbb{P}\Big( \sum_{s \neq r} Y_s = \lfloor R/2 \rfloor \Big).$$

To establish our main bound, detailed in Appendix E, we interpret $U(\hat{Y}_R^{\phi^{\mathrm{MV}}})$ as a difference of a function $F$ depending on $l_i(a) := \mathbb{P}(\tilde{Y}_r = 1 \mid A = a)$. We apply the Mean Value Theorem and use Baillon et al. (2015)'s Theorem to get a sharp bound on $\partial_{p_i} F$ after invoking Lemma 3.5. While Theorem 3.4 relies on annotator skills, this bound depends only on their individual parity gap and their variance.

For the rest, let $V_R(a) := \sum_{i=1}^{R} l_i(a)\big(1 - l_i(a)\big)$.

**Proposition 3.6.** *Let $\hat{Y}_R^{\phi^{\mathrm{MV}}}$ denote the Majority Vote and $(\tilde{Y}_r)_{r \in [R]}$ be the noisy labels. Then one can find $\eta \in ]0, \frac{1}{2}[$ such that for any $R \geq 1$:*

$$\Delta_{\mathrm{DP}}(\hat{Y}_R^{\phi^{\mathrm{MV}}}) \leq \epsilon(R) \sum_{r=1}^{R} \Delta_{\mathrm{DP}}(\tilde{Y}_r), \quad (10)$$

*where $\eta = \max_{\lambda \geq 0} \sqrt{2\lambda} e^{-2\lambda} \sum_{k=0}^{\infty} \left( \frac{\lambda^k}{k!} \right)^2 \approx 0.4688$, and $\varepsilon(R) := \eta \left( \min\left\{ \sqrt{V_R(0)}, \ \sqrt{V_R(1)} \right\} \right)^{-1}$.*

If the crowd exhibits systematic bias, i.e. biases are aligned such that $\Delta_{\mathrm{DP}}(\tilde{Y}_r) \approx \delta > 0$, the aggregate bias scales as

$\mathcal{O}(\sqrt{R})$ to 1, the maximum value of the bias. This suggests that without correction, Majority Vote acts as a bias accumulator rather than a filter in the small crowd regime, confirming the necessity of the regularization mechanisms proposed in Section 4.

# 4. The FairCrowd Post-Processing Algorithm

In this section, to align with the $\epsilon$-fair classification framework, we define $W = (\tilde{Y}_{1:R}, X) \in \mathcal{W} = \{0,1\}^R \times \mathcal{X}$.

Under this formulation, $\phi : \mathcal{W} \times \{0,1\} \mapsto \{0,1\}$ can be viewed as a binary classifier, and we write $\phi(w,a) = \mathbb{I}\{\hat{\Phi}_1(w,a) \geq 1/2\}$, where $\hat{\Phi}$ is an estimator of the posterior probability $P_1^*(w,a)$, with $P_k^*(w,a) = \mathbb{P}(Y = k \,|\, W = w, A = a)$ for any $(w,a) \in \mathcal{W} \times \{0,1\}$. For instance, $\hat{\Phi}_1 = \sum_{i=1}^R \tilde{Y}_i / R$ for the Majority Vote and $\hat{\Phi}_1(w,a) = \mathbb{P}(Y = 1 \,|\, W = w, A = a)$ for the Bayesian Vote. We propose a post-processing algorithm that regularizes any given classifier $\phi$ so that its prediction $\hat{Y}_R^\phi$ satisfies $\epsilon$-fairness with respect to demographic parity, while preserving accuracy as much as possible.

More precisely, we solve the following problem:

$$\arg \min_{\phi \in \mathcal{G}_\epsilon} \mathcal{R}, \quad \mathcal{R}(\phi) = \mathbb{P}(\hat{Y}^\phi \neq Y), \qquad (11)$$

where $\mathcal{G}_\epsilon$ is the set of random classifier $\phi$ such that $\hat{Y}^\phi \sim$ Bernoulli$(\phi(W,A))$ satisfies $\Delta_{\mathrm{DP}}(\hat{Y}^\phi) \leq \epsilon$. Note that in the previous section the considered aggregation methods $\phi$ were deterministic since $\phi(W,A) \in \{0,1\}$. Here, random classifier are used to get existence of solution of (11).

This problem was previously studied in Denis et al. (2024), but under the assumption that the maps $t \mapsto \mathbb{P}(P_1^*(W,A) \leq t \,|\, A = a)$ are continuous for all $a \in \{0,1\}$. In the crowdsourcing setting, however, this assumption is overly restrictive, since $(W, A)$ may take only finitely many values, and the same holds for $P_1^*(W,A)$. This situation is especially common when no continuous non-sensitive features $X$ are available as $W = \tilde{Y}_{1:R} \in \{0,1\}^R$, which is often the case in crowdsourcing applications (Raykar et al., 2010; Ibrahim et al., 2025). .

Next, we solve (11) in the binary case, $Y \in \{0,1\}$, without any assumption, by following the approach of Denis et al. (2024). The $K$-class generalization is in Appendix A.

**Theorem 4.1.** *Denoting by $s_a = 2a-1$ and $\pi_a = \mathbb{P}(A = a)$ for any $a \in \{0,1\}$, the solution of (11) is $\phi_{\beta^*}^*$ given as:*

$$\phi_{\beta^*}^*(w,a) = \begin{cases} 1, & \text{if } P_1^*(w,a) > \frac{\pi_a + s_a \beta^*}{2\pi_a}, \\ 0, & \text{if } P_1^*(w,a) < \frac{\pi_a + s_a \beta^*}{2\pi_a}, \\ \omega_a, & P_1^*(w,a) = \frac{\pi_a + s_a \beta^*}{2\pi_a}, \end{cases}$$

*where $\beta^* = \arg \min_{\beta \in \mathbb{R}} \mathcal{M}(\beta)$ with $\mathcal{M}(\beta) = \mathcal{L}(\beta) + \epsilon|\beta|$,*

$$\mathcal{L}(\beta) = \sum_{a=0}^1 \mathbb{E}_{W|A=a} \left( \max_{k \in \{0,1\}} \pi_a P_k^*(W,a) - \frac{\beta}{2} s_a s_k \right),$$

*and $(\omega_a)_{a \in \{0,1\}}$ are defined as the unique solution of*

$$\epsilon^* = \mathbb{P}(\phi_{\beta^*}^*(W,A) = 1 | A = 1) - \mathbb{P}(\phi_{\beta^*}^*(W,A) = 1 | A = 0), \qquad (12)$$

*with minimal norm $|\omega_0| + |\omega_1|$,*
*where $\epsilon^* = \epsilon (\mathbb{I}_{\beta^* \neq 0} \operatorname{sign}(\beta^*) + \xi \mathbb{I}_{\beta^* = 0})$ and $\xi \in [-1, 1]$.*

Note that if $\beta^* = 0$, we recover the classical Bayes classifier. Theorem 4.1 states that, to solve (11), one should apply the minimal adjustment to the decision boundary of the Bayes-optimal aggregation rule $\phi^*$ in order to satisfy the fairness constraints in (12).

The Theorem is derived by using a min-max argument and by working cautiously on the optimality conditions. Especially, constants $\omega_a$ appear because of a sub-gradient taken at a point of non-differentiability of $\mathcal{M}$. The proof in the multi-class follows the same idea.

**Implementation.** Theorem 4.1 offers a recipe to design an optimal $\epsilon$-fair classifier. Given a dataset $\mathsf{W} = (W_i^0) \cup (W_i^1)$ split in two groups, respectively, of size $N_0, N_1$, according to the sensitive feature $A = 0, 1$. $\pi_a$ is estimated by $\hat{\pi}_a = N_a / (N_0 + N_1)$ and $P_k^*(w,a)$ by the crowdsourced aggregation method probabilities $\hat{\Phi}_k(w,a)$. $\mathcal{L}$ is approximated as follows:

$$\hat{\mathcal{L}}(\beta) = \sum_{a=0}^1 \frac{1}{N_a} \sum_{i=0}^{N_a} \operatorname{soft}_c \max_{k \in \{0,1\}} (\hat{\pi}_a \hat{\Phi}_k(W_i^a, a) - \frac{\beta}{2} s_a s_k), \qquad (13)$$

where $0 < c \ll 1$ and for any $(a_k)_{k \in \mathsf{J}}$,

$$\operatorname{soft}_c \max_{k \in \mathsf{J}} a_k = \sum_{k \in \mathsf{J}} \frac{\exp(a_k/c)}{\sum_{j \in \mathsf{J}} \exp(a_j/c)} a_k.$$

$\beta^*$ is estimated by minimizing $\hat{\mathcal{M}} : \beta \mapsto \hat{\mathcal{L}} + \epsilon |\cdot|$. In our experiments we used sequential quadratic programming.

As $(\omega_0, \omega_1) \in [0,1]^2$ can be estimated using a grid-search to solve (12) and using $(|P_1^*(w,a) - \frac{\pi_a + s_a \beta^*}{2\pi_a}| < \delta)$ instead of $(P_1^*(w,a) = \frac{\pi_a + s_a \beta^*}{2\pi_a})$ with $0 < \delta \ll 1$ to take into account numerical approximation.

Sometimes, especially when using EM (Dawid & Skene, 1979), crowdsourced aggregation methods generate posterior estimates $\hat{\Phi}_k(w,a)$ that concentrate too much around 0 or 1 to make the optimization problem (13) accurate, such that numerical optimization methods return $\beta^* = 0$ since $\hat{\mathcal{M}}$ is nearly constant. To avoid numerical errors, denoting by $p_i = \hat{\Phi}_1(W_i, a_i)$, we apply the following preprocessing

**Algorithm 1** FairCrowd

**Input:** $\hat{\Phi}_k(w, a)$: Estimated posterior probabilities; $\epsilon$: fairness gap; $\hat{\pi}_a$: estimated group probabilities $\hat{\pi}_a \approx \mathbb{P}(A = a)$ for any $a \in \{0, 1\}$.
**Output:** $\phi_{\beta^*}^*$ defined in Theorem 4.1.

**Step 0:** Apply the trick (14) to make $(\hat{\Phi}_k(w, a))$ less concentrated around 0 and 1.
**Step 1:** Minimize $\beta \mapsto \hat{\mathcal{M}}(\beta) = \hat{\mathcal{L}}(\beta) + \epsilon|\beta|$ with sequential quadratic programming, where $\hat{\mathcal{L}}$ is defined in (13) with the softmax parameter $c = 10^{-4}$.
**Step 2:** Using a grid-search on $(\omega_0, \omega_1)$ solve (17) using $(|P_1^*(w, a) - \frac{\pi_a + s_a \beta^*}{2\pi_a}| < \delta)$, with $\delta = 10^{-5}$, instead of $(P_1^*(w, a) = \frac{\pi_a + s_a \beta^*}{2\pi_a})$.
**Return:** $\phi_{\beta^*}^*$ defined in Theorem 4.1.

step:

$$\hat{\Phi}_1(W_i, a_i)^{\text{pre}} = f(p_i),$$
$$\Phi_0(W_i, a_i)^{\text{pre}} = 1 - f(p_i) \tag{14}$$
$$f(p_i) = \mathbb{I}_{[0,1-\alpha]}(p_i)p_i + \mathbb{I}_{[1-\alpha,1]}(p_i)k(p_i)\eta,$$

where $\alpha = 0.04$, $\eta = 1/|\mathsf{I}_\alpha|$, $\mathsf{I}_\alpha = \{p_j : p_j \in [1-\alpha, 1]\}$ and $k(p_i)$ is the index of $p_i$ in $\mathsf{I}_\alpha$ if $\mathsf{I}_\alpha$ is sorted by increasing order.

The recipe is summarized in Algorithm 1.

In practice, not all annotators label every task. Nevertheless, the method remains effective, since no assumptions are imposed on $W$. What is essential for achieving good accuracy is obtaining a reasonable approximation $\hat{\Phi}_k \approx P_k^*$ of the posterior. Notably, even if this approximation is poor, FairCrowd still returns an $\epsilon$-fair randomized classifier.

**Multiclass case.** When $Y$ is multiclass (Theorem A.1), the dimension of the problem increases to $d$ such that $\beta \in \mathbb{R}^d$, compared to $2d$ in Denis et al. (2024). In the multiclass case, $\beta$ can still be estimated by minimization, but the generalized $\omega_a$ are burdensome to express, such that it is better in practice to use the trick given by Denis et al. (2024), by artificially adding some noise to the estimated posterior probability $\hat{\Phi}$ to satisfy their continuity assumption (Denis et al., 2024, Assumption 2.1). While our multiclass generalization Theorem A.1 is not used in our implementation, it still offers a deeper understanding of the structure of the problem in the discrete case, and the proof reveals the link with the subdifferential of $\mathcal{L}$.

# 5. Experiments

First, theoretical results are verified empirically on synthetic data. Second, our suggested fairness post-processing method called FairCrowd (FC) is evaluated on both synthetic and real-world datasets against the state-of-the-art (Li et al., 2020).

**Crowdsourcing label integration algorithms.** Since both fairness post-processing algorithm can be applied to any crowdsourced aggregation method, we focus on three widely used and well-understood baselines: Majority Vote (5) (Maj), Bayesian optimal aggregation (4) (Bayes,$\phi^*$) (Nitzan & Paroush, 1982; Snow et al., 2008), and the Dawid-Skene model (DS,$\phi^{DS}$) (Dawid & Skene, 1979). $\phi^*, \phi^{DS}$ are defined using the same formula (4), developed in Appendix B with numerical details, but with different estimation for the annotator confusion probabilities $\pi_{k,y}^{(r)}(a) = \mathbb{P}(\tilde{Y}_r = k | Y = y, A = a)$. Note that the Bayesian optimal aggregation $\phi^*$ estimates $\pi_{k,y}^{(r)}(a)$ by counting, using ground-truth data, while the Dawid-Skene model approximates the Bayesian optimal aggregation without ground-truth, using an EM-based approach.

## 5.1. Convergence illustration on synthetic data

To assess the theoretical results, we generate a synthetic dataset with $\mathbb{P}(A = 1) = \frac{1}{2}$ and $\mathbb{P}(Y = 1 | A = a) = \frac{1}{2}$. No features are generated, *i.e.* $\mathcal{X} = \emptyset$. The annotators' skills $P_r(a) = \mathbb{P}(\tilde{Y}_r = Y | A = a)$ are sampled independently according to $\mathcal{K}_a$, namely:

$$P_r(a) \overset{i.i.d.}{\sim} \mathcal{K}_a = \mathcal{U}([0.5\,\mathbb{I}_0(a) + 0.6\,\mathbb{I}_1(a), 1]).$$

In Figure 2, we empirically evaluate the difference $\Delta_{\text{DP}}(\hat{Y}^\phi) - \Delta_{\text{DP}}(Y)$ for $R \in \{3, 5, 8, 10, 15, 20, 40\}$ and $\phi \in \{\phi^*, \phi^{MV}, \phi^{DS}\}$.

We clearly observe in Figure 2-(a) the convergence $\Delta_{\text{DP}}(\hat{Y}^\phi) \to \Delta_{\text{DP}}(Y)$ as the number of annotators $R \to +\infty$, which is expected since all the assumptions of Theorem 3.4 are satisfied, as most annotators verify $p_r > 1/2 + \epsilon$. However, Majority Vote amplifies the bias in the small-crowd regime ($R < 15$), which is again consistent with Theorem 3.6.

In the case where $\mathcal{K}_a = \mathcal{U}([0.2\mathbb{I}_0(a) + 0.1\mathbb{I}_1(a), 0.6])$, illustrated in Figure 2-(b), the annotators are globally adversarial. As a consequence, the assumptions of Theorem 3.4 are not satisfied for Majority Vote: convergence does not hold for $\phi^{MV}$, while it still holds for $\phi^*$ and, to some extent, for $\phi^{DS}$.

Finally, in the case where $\mathcal{K}_a = \mathcal{U}([0.49, 0.51])$, illustrated in Figure 2-(c), the annotators convey almost no information. As a result, the assumptions of Theorem 3.4 are not satisfied for the Bayesian aggregation $\phi^*$ due to estimation error: convergence does not hold for $\phi^*$. Nevertheless, convergence is still observed empirically for $\phi^{MV}$ and $\phi^{DS}$, since the annotators' labels remain balanced.

## 5.2. Comparison of $\epsilon$-fair label integration methods

We first introduce the datasets, the competing methods, and the comparison methodology, before presenting the results

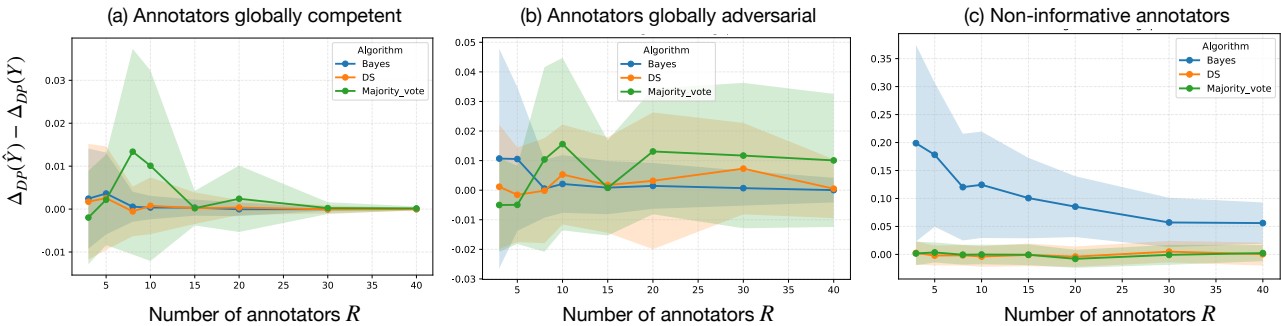

*Figure 2.* Convergence of the difference of demographic parity gap $\Delta_{\mathrm{DP}}(\hat{Y}^{\phi}) - \Delta_{\mathrm{DP}}(Y)$ according to the number of annotators $R$ for any $\phi \in \{\phi^*, \phi^{\mathrm{MV}}, \phi^{DS}\}$ in the case where annotators (a) are globally competent $\mathcal{K}_a = \mathcal{U}([0.5\,\mathbb{I}_0(a) + 0.6\,\mathbb{I}_1(a), 1])$, (b) are globally adversarial $\mathcal{K}_a = \mathcal{U}([0.2\mathbb{I}_0(a) + 0.1\mathbb{I}_1(a), 0.6])$, (c) convey nearly no information.

and their implications.

**Synthetic dataset.** We generate a synthetic dataset with $\mathbb{P}(A = 1) = 0.6$ and $\mathbb{P}(Y = 1 \,|\, A = a) = \frac{1}{2} + (2a - 1)\,0.1$. No features are generated, *i.e.* $\mathcal{X} = \emptyset$. The annotators' skills $P_r(a) = \mathbb{P}(\tilde{Y}_r = Y \,|\, A = a)$ are sampled independently according to $\mathcal{K}_a$, namely $P_r(a) \overset{i.i.d.}{\sim} \mathcal{K}_a = \mathcal{U}([0.5\,\mathbb{I}_0(a) + 0.6\,\mathbb{I}_1(a), 1])$, as in Figure 2-(a). The dataset consists of $N = 2000$ tasks, each labeled by 5 annotators uniformly sampled without replacement from a pool of $R = 100$ annotators.

**Real datasets.** We rely on the same benchmark real datasets as in Lazier et al. (2023), and additionally include an imbalanced dataset Kennedy et al. (2020). For the unbalanced dataset see appendix H for detailed plot.

*Crowd Judgment (Dressel & Farid, 2018).* This dataset contains labels provided by annotators from a major crowdsourcing platform for $1,000$ cases drawn from the well-known COMPAS recidivism prediction task, based on defendant profiles from Broward County, Florida. Groups of 20 annotators evaluated the same set of 50 defendants. The task consists of predicting whether a defendant will reoffend within two years. The sensitive attribute $A$ represents skin color, coded as $A = 1$ for Black and $A = 0$ for non-Black.

*Jigsaw Toxicity (jig, 2023).* This dataset originates from the Civil Comments platform and consists of online comments labeled for toxicity by multiple human raters. We use a subset of $5,000$ comments labeled for toxicity, obscenity, threats, insults, and hate. Annotators are unevenly distributed across comments, resulting in a heavy-tailed distribution of the number of labels per comment, with a median of 4 and a standard deviation of $75.64$. The sensitive feature $A$ is 1 if the comment mentions a discriminated group and 0 otherwise.

*Measuring Hate Speech (Kennedy et al., 2020).* This dataset, consists of $50,000$ social media comments sourced from YouTube, Twitter, and Reddit, each rated by multiple Amazon Mechanical Turk workers from a pool of roughly $10,000$ annotators. The task consists of predicting whether a comment qualifies as hate speech. The sensitive feature A indicates whether the comment targets a specific gender identity group. The distribution of $Y$ labels is imbalanced since hateful content represents only a small fraction of online communication.

**Competitors.** We compare our method against the in-processing (FairTD) and post-processing (Post_TD), both from Li et al. (2020). For each annotator $i$ and sensitive feature value $A = a$, FairTD estimates a bias term $b_i^a$ and an accuracy parameter $\sigma_i^2 > 0$ using a linear mixed-effects model. Denoting by $Y_{i,j}$ the label provided by annotator $i$ for task $j$, and by $Y_j$ the true label of task $j$, the model is given by:

$$\tilde{Y}_{i,j} = Y_j + b_i^a + \epsilon_{i,j}, \quad \epsilon_{i,j} \sim \mathcal{N}(0, \sigma_i^2). \quad (15)$$

The bias is then corrected for annotators whose labels violate demographic parity, while preserving the positive bias of other annotators. Consensus labels are predicted via a weighted aggregation of annotator labels, using the estimated accuracies $\sigma_i^2$ as weights. Bias estimation and positive-bias selection are performed iteratively through a sampling-based procedure, which terminates once the global demographic parity (DP) constraint is satisfied and the parameter estimates have converged (see Algorithm 1 in Li et al. (2020)). Note that FairTD is not a post-processing method, hence has a reduced application scope.

Given the predictions of a crowdsourced aggregation algorithm, Post_TD applies a post-processing step that flips labels conditioned on the sensitive feature in order to satisfy the global DP constraint. This procedure adapts the massaging method introduced in Kamiran & Calders (2012).

Each crowdsourcing method (Maj, Bayes, DS) is evaluated with a fairness post-processing step using either our approach (FC) or Post_TD.

**Methodology.** We compare the performance of the crowd-

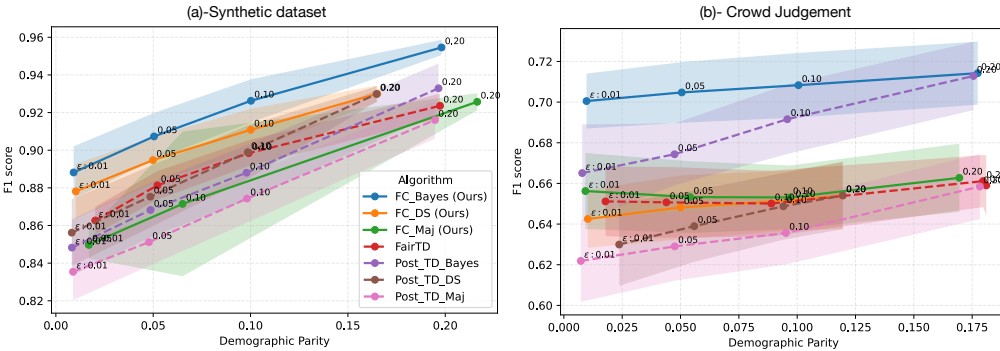

*Figure 3.* Performance comparison on the Synthetic dataset (a) and Crowd Judgement dataset (b). Solid lines correspond to our method (FC), while dashed lines indicate competing approaches. Shaded regions denote the variance across 10 independent runs using different test set (60%).

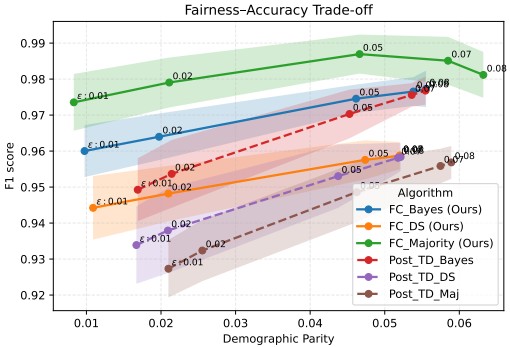

*Figure 4.* Performance comparison on the Jigsaw dataset. Solid lines correspond to our method (FC), while dashed lines indicate competing approaches. Shaded regions denote the variance across 10 independent runs using different test set (60%).

sourced aggregation by plotting the F1-score of the different method according to the Global DP for different choice of $\epsilon \in \{0.01, 0.05, 0.1, 0.2\}$ as parameter of fairness methods (FairTD, Post_TD,FC). On the Jigsaw dataset, we select $\epsilon \in \{0.01, 0.02, 0.05, 0.07, 0.08\}$ because Jigsaw offers lower demographic parity gaps than the other datasets. While for Hate Speech data set $\epsilon \in \{0.05, 0.1, 0.15, 0.2\}$. FairTD is not presented on Jigsaw and HateSpeech datasets as it does not terminate within 30 hours using the implementation given in (Li et al., 2020), which is to be compared with our method (FC) and Post_TD that terminate in less than 2 seconds. The F1-score is evaluated on a test set representing 60% of the dataset which is 10 times resampled to analyze performance variability represented by the shaded area in the different figures.

**Results.** First, as shown in Figures 3 and 4, fairness post-processing (FC, Post_TD) is effective, even if Post_TD might be innaccurate on Jigsaw for $\epsilon = 0.01$. The same holds for FairTD: the values of $\epsilon$ and the Demographic Parity gap align, except when $\epsilon > \Delta_{\mathrm{DP}}(\hat{Y})$, which is expected since the parity constraint is not restrictive.

Second, across all experiments and crowdsourcing methods, our approach (FC) outperforms Post_TD, particularly for small values of $\epsilon$. Note that Post_TD does not utilize the probability estimates $\hat{\Phi}$ produced by the crowdsourcing methods; it only considers the predictions $\hat{Y}$, discarding potentially important information, especially as $\epsilon$ decreases. On the Measuring Hate Speech dataset, which is highly imbalanced, we observe larger variability across resampled test sets. Nevertheless, FC remains competitive with Post_TD across the considered fairness constraints, suggesting that the proposed post-processing remains stable even under severe class and group imbalance see Figure 5.

Third, applying our method to Majority Vote (FC_Maj) consistently performs as well as or better than FairTD, suggesting that the aggregation component in FairTD offers no advantage over a simple Majority Vote.

Fourth, Bayesian optimal aggregation estimated using the ground-truth (Bayes) performs best on the synthetic and Crowd Judgement datasets, but not on the Jigsaw dataset, where Majority Vote outperforms it. This can be explained by the fact that the annotators in Jigsaw are highly reliable, making methods like Bayes or DS, which rely on empirical estimation of probabilities, less effective, particularly when each annotator labels only a few tasks.

Fifth, although enforcing $\epsilon$-fairness generally reduces the F1-score, this is not always the case. For instance, FC applied to Majority Vote (FC_Maj) on the Crowd Judgement dataset remains stable regardless of the choice of $\epsilon$.

## 6. Conclusion

In this work, we developed a unified theoretical and algorithmic framework for fairness in crowdsourced label aggregation. We derived the first sharp non-asymptotic bounds on the fairness gap of Majority Vote and Bayesian aggregation, showing exponential convergence to ground-truth fairness under mild, interpretable conditions on annotator

quality, and revealing bias amplification in the small-crowd regime for Majority Vote. We generalized the $\varepsilon$-fairness post-processing framework of Denis et al. (2024) to binary and multi-class crowdsourcing with discrete inputs. We then proposed *FairCrowd*, a practical post-processing algorithm that enforces $\varepsilon$-fairness for arbitrary aggregation rules and achieves state-of-the-art performance on synthetic and real-world benchmarks. In particular, FairCrowd applied to Majority Vote is computationally efficient and performs well even without prior information. Overall, our results bridge theory and practice and provide principled tools for fair and reliable crowdsourcing systems. In our experiments, we do not model interactions between sensitive features and inputs when estimating confusion matrices. Addressing feature-dependent annotator behavior is left for future work, as it raises substantial challenges and potential identifiability issues (Ibrahim et al., 2025; Nguyen et al., 2024).

## Impact Statement

Crowdsourcing is widely used to gather large-scale labeled data, but the variability in annotator quality and potential biases can propagate unfairness into downstream machine learning models. In this work, we introduce a principled approach that leverages annotator-level probability estimates to enforce fairness through post-processing, improving demographic parity without sacrificing predictive performance. Our method unifies aggregation and fairness adjustment, outperforming standard techniques such as Majority Vote or naive fairness post-processing across diverse datasets. Beyond empirical gains, this work provides theoretical insights into how annotator heterogeneity interacts with fairness constraints, offering guidance for the design of reliable and equitable crowdsourced data pipelines. By bridging crowdsourcing, aggregation, and fairness, our approach lays a foundation for more responsible machine learning systems that remain robust even when labels come from imperfect human sources.

## Acknowledgements

We thank SNCF and RTE for their industrial support and for the real-world issues that gave rise to this research problem. We also thank Simone Lazier for sharing part of the code used in our experimental section. The authors affiliated with Centre Borelli acknowledge the support of the Industrial Analytics and Machine Learning (IdAML) Chair hosted at ENS Paris-Saclay, Université Paris-Saclay.

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

## A. Multi-class optimal $\epsilon$-fair aggregation by generalizing (Denis et al., 2024)

Here, there are $K$ classes, i.e. $Y \in \{1, K\}$, such that a random classifier $g$ outputs a value $g(w, a)$ in the simplex $\Delta = \{(\omega_k) \in \mathbb{R}_{\geq 0}^K : \sum_{k=1}^K \omega_k = 1\}$ and the prediction is $\hat{Y} \sim \text{Multinomial}(g(w, a))$.

**Theorem A.1.** *Denoting by $s_a = 2a - 1$ and $\pi_a = \mathbb{P}(A = a)$ for any $a \in \{0, 1\}$, the solution of* (11) *is $\phi_{\beta^*}^*$ given as*

$$
\phi_{\beta^*}^*(w, a) = \begin{cases} (\mathbb{I}_{\{k\}}(i))_{i \in \{1, ..., K\}}, & \text{if } \{k\} = \arg\max_{k \in \{1, ..., K\}} (\pi_a P_k^*(w, a) - s_a \beta_k^*) \quad j \neq k \\ (\omega_{a,i}^{\mathsf{J}} \mathbb{I}_{\mathsf{J}}(i))_{i \in \{1, ..., K\}}, & \text{if } \arg\max_{k \in \{1, ..., K\}} (\pi_a P_k^*(w, a) - s_a \beta_k^*) = \mathsf{J}, \quad \mathsf{J} \subset \{1, ..., K\}, |\mathsf{J}| > 1 \end{cases}
$$

*where $(\beta_k^*)_{k \in \{1, ..., K\}} = \beta^* = \arg\min_{\beta \in \mathbb{R}^d} \mathcal{M}(\beta)$ with $\mathcal{M}(\beta) = \mathcal{L}(\beta) + \epsilon |\beta|$,*

$$
\mathcal{L}(\beta) = \sum_{a=0}^1 \mathbb{E}_{W|A=a} \left( \max_{k \in \{1, ..., K\}} \pi_a P_k^*(W, a) - \beta_k s_a \right) \tag{16}
$$

*and $(\omega_{a,k}^{\mathsf{J}})$ are defined as solution of*

$$
\epsilon_k^* = \mathbb{P}(\phi_{\beta^*}^*(W, A) = k | A = 1) - \mathbb{P}(\phi_{\beta^*}^*(W, A) = k | A = 0) \tag{17}
$$

*with $\epsilon_k^* = \epsilon \left( \mathbb{I}_{\beta_k^* \neq 0} \text{sign}(\beta_k^*) + \xi \mathbb{I}_{\beta_k^* = 0} \right)$ and $\xi_k \in [-1, 1]$.*

The proof is given in Appendix G.

## B. Expression of the Bayesian optimal aggregation $\phi^*$.

**Proposition B.1.** *The Bayes optimal classifier is given by*

$$
\phi^*(\tilde{Y}, X, A) = \mathbf{1} \left\{ \mathbb{P}(Y = 1 | \tilde{Y}, X, A) \geq \tfrac{1}{2} \right\},
$$

*where for any $\tilde{y} = (\tilde{y}_1, ..., \tilde{y}_R) \in \{0, 1\}^R$, $a \in \{0, 1\}$, and $x \in \mathcal{X}$, with $\pi_{y', y}^{(r)}(x, a) = \mathbb{P}(\tilde{Y}_r = y' | Y = y, A = a, X = x)$, we have:*

$$
\mathbb{P}(Y = 1 | \tilde{Y} = \tilde{y}, X = x, A = a) = \frac{1}{1 + \Pi(\tilde{y}, x, a)},
$$

$$
\Pi(\tilde{y}, x, a) = \frac{\mathbb{P}(Y = 0 | X = x, A = a)}{\mathbb{P}(Y = 1 | X = x, A = a)} \prod_{r=1}^R \left( \frac{\pi_{1,0}^{(r)}(x, a)}{\pi_{1,1}^{(r)}(x, a)} \right)^{\tilde{y}_r} \left( \frac{\pi_{0,0}^{(r)}(x, a)}{\pi_{0,1}^{(r)}(x, a)} \right)^{1 - \tilde{y}_r}.
$$

*Proof.* We derive the posterior probability using Bayes' theorem and the conditional independence of the $\tilde{Y}_r$ given $Y$, $A$, and $X$.

By Bayes' theorem:

$$
\mathbb{P}(Y = 1 | \tilde{Y} = \tilde{y}, X = x, A = a) = \frac{\mathbb{P}(\tilde{Y} = \tilde{y} | Y = 1, X = x, A = a) \mathbb{P}(Y = 1 | X = x, A = a)}{\mathbb{P}(\tilde{Y} = \tilde{y} | X = x, A = a)}.
$$

The denominator can be expressed via the law of total probability:

$$
\mathbb{P}(\tilde{Y} = \tilde{y} | X = x, A = a) = \sum_{y \in \{0, 1\}} \mathbb{P}(\tilde{Y} = \tilde{y} | Y = y, X = x, A = a) \mathbb{P}(Y = y | X = x, A = a).
$$

Under the conditional independence assumption given $Y$, $A$, and $X$:

$$
\mathbb{P}(\tilde{Y} = \tilde{y} | Y = y, X = x, A = a) = \prod_{r=1}^R \mathbb{P}(\tilde{Y}_r = \tilde{y}_r | Y = y, X = x, A = a).
$$

Set:

$$p_1(x, a) := \mathbb{P}(Y = 1 \mid X = x, A = a),$$
$$p_0(x, a) := \mathbb{P}(Y = 0 \mid X = x, A = a) = 1 - p_1(x, a).$$

For each $r$, define:

$$\alpha_r(\tilde{y}_r, x, a) = \mathbb{P}(\tilde{Y}_r = \tilde{y}_r \mid Y = 1, X = x, A = a),$$

$$\beta_r(\tilde{y}_r, x, a) = \mathbb{P}(\tilde{Y}_r = \tilde{y}_r \mid Y = 0, X = x, A = a).$$

Then the posterior can be written as:

$$\mathbb{P}(Y = 1 \mid \tilde{Y} = \tilde{y}, X = x, A = a) = \frac{p_1(x, a) \prod_{r=1}^{R} \alpha_r(\tilde{y}_r, x, a)}{p_1(x, a) \prod_{r=1}^{R} \alpha_r(\tilde{y}_r, x, a) + p_0(x, a) \prod_{r=1}^{R} \beta_r(\tilde{y}_r, x, a)}$$
$$= \frac{1}{1 + \frac{p_0(x,a)}{p_1(x,a)} \prod_{r=1}^{R} \frac{\beta_r(\tilde{y}_r, x, a)}{\alpha_r(\tilde{y}_r, x, a)}}.$$

Now, we express the ratios $\frac{\beta_r(\tilde{y}_r, x, a)}{\alpha_r(\tilde{y}_r, x, a)}$ in terms of the parameters $\pi_{y',y}^{(r)}(x, a)$.

Note that:

$$\pi_{1,1}^{(r)}(x, a) = \mathbb{P}(\tilde{Y}_r = 1 \mid Y = 1, X = x, A = a) = \alpha_r(1, x, a),$$

$$\pi_{1,0}^{(r)}(x, a) = \mathbb{P}(\tilde{Y}_r = 1 \mid Y = 0, X = x, A = a) = \beta_r(1, x, a),$$

$$\pi_{0,1}^{(r)}(x, a) = \mathbb{P}(\tilde{Y}_r = 0 \mid Y = 1, X = x, A = a) = \alpha_r(0, x, a),$$

$$\pi_{0,0}^{(r)}(x, a) = \mathbb{P}(\tilde{Y}_r = 0 \mid Y = 0, X = x, A = a) = \beta_r(0, x, a).$$

Therefore,

$$\frac{\beta_r(1, x, a)}{\alpha_r(1, x, a)} = \frac{\pi_{1,0}^{(r)}(x, a)}{\pi_{1,1}^{(r)}(x, a)}, \quad \frac{\beta_r(0, x, a)}{\alpha_r(0, x, a)} = \frac{\pi_{0,0}^{(r)}(x, a)}{\pi_{0,1}^{(r)}(x, a)}.$$

We can combine these two cases using the indicator form:

$$\frac{\beta_r(\tilde{y}_r, x, a)}{\alpha_r(\tilde{y}_r, x, a)} = \left( \frac{\pi_{1,0}^{(r)}(x, a)}{\pi_{1,1}^{(r)}(x, a)} \right)^{\tilde{y}_r} \left( \frac{\pi_{0,0}^{(r)}(x, a)}{\pi_{0,1}^{(r)}(x, a)} \right)^{1 - \tilde{y}_r}.$$

So

$$\mathbb{P}(Y = 1 \mid \tilde{Y} = \tilde{y}, X = x, A = a) = \frac{1}{1 + \frac{p_0(x,a)}{p_1(x,a)} \prod_{r=1}^{R} \left( \frac{\pi_{1,0}^{(r)}(x,a)}{\pi_{1,1}^{(r)}(x,a)} \right)^{\tilde{y}_r} \left( \frac{\pi_{0,0}^{(r)}(x,a)}{\pi_{0,1}^{(r)}(x,a)} \right)^{1 - \tilde{y}_r}}.$$

Defining $\Pi(\tilde{y}, x, a)$ as in equation (2) completes the proof. $\qquad\square$

**Experimental setting:** We assume no feature dependence: $\pi_{y',y}^{(r)}(x, a) = \pi_{y',y}^{(r)}(a)$. All probabilities for the method **Bayes** $\phi*$ are estimated empirically by counting using ground-truth labels. For **DS**, the probabilities are estimated using $n = 20$ iterations of EM algorithms by initializing $\hat{\mathbb{P}}(Y = 1 \mid X = x, A = a) = 1/2$ and assuming the one-coin model with uniform skills of 0.7, i.e., $\hat{\pi}_{y,y}^{(r)}(a) = 0.7$ for any $(y, a) \in \{0, 1\}^2$.

## C. From punctual fairness to global fairness

Before giving the explicit counter exemple we recall a useful lemma about conditional probability and total probability theorem:

**Lemma C.1.** *Let $A, X, Y$ be three random variables taking values in $\{0, 1\}$. One has:*

$$\mathbb{P}(Y = 1 \mid A = 1) = \sum_{x \in \{0,1\}} \mathbb{P}(Y = 1 \mid X = x, A = 1)\mathbb{P}(X = x \mid A = 1) \tag{18}$$

*Proof.* First of all notice that one can write:

$$(Y = 1 \cap A = 1) = \bigcup_{x \in \{0,1\}} (Y = 1 \cap A = 1 \cap X = x)$$

$$\implies \mathbb{P}(Y = 1 \cap A = 1) = \sum_{x \in \{0,1\}} \mathbb{P}(Y = 1 \cap X = x \cap A = 1).$$

By definition: $\quad \mathbb{P}(Y = 1 \mid A = 1) = \dfrac{\mathbb{P}(Y = 1 \cap A = 1)}{\mathbb{P}(A = 1)}$

$$= \frac{\sum_{x \in \{0,1\}} \mathbb{P}(Y = 1 \mid X = x, A = 1)\mathbb{P}(X = x \cap A = 1)}{\mathbb{P}(A = 1)}$$

$$= \sum_{x \in \{0,1\}} \mathbb{P}(Y = 1 \mid X = x, A = 1)\underbrace{\frac{\mathbb{P}(X = x \cap A = 1)}{\mathbb{P}(A = 1)}}_{\mathbb{P}(X=x \mid A=1)}$$

$$= \sum_{x \in \{0,1\}} \mathbb{P}(Y = 1 \mid X = x, A = 1)\mathbb{P}(X = x \mid A = 1).$$

$\square$

**Proposition C.2.** *Let $\tilde{Y}$ be a binary random variable such that the annotator is perfectly pointwise fair, i.e. $\mathcal{U}(\tilde{Y}, x) = 0$ for all $x \in \mathcal{X}$. Then one can find $\tilde{Y}$ that achieves:*

$$\Delta_{\mathrm{DP}}(\tilde{Y}) \neq 0.$$

*Proof.* We construct an explicit example with binary $A \in \{0, 1\}$ and binary $X \in \{0, 1\}$. Define the joint distribution as follows:

- $\mathbb{P}(A = 1) = \frac{1}{2}, \mathbb{P}(A = 0) = \frac{1}{2}$.
- $\mathbb{P}(X = 1 \mid A = 1) = \frac{3}{4}, \mathbb{P}(X = 1 \mid A = 0) = \frac{1}{4}$.
- $\mathbb{P}(\tilde{Y} = 1 \mid X = 0, A = a) = 0.2$ for all $a \in \{0, 1\}$.
- $\mathbb{P}(\tilde{Y} = 1 \mid X = 1, A = a) = 0.8$ for all $a \in \{0, 1\}$.

By construction, for any $x \in \{0, 1\}$:

$$Q_1(x) = \mathbb{P}(\tilde{Y} = 1 \mid X = x, A = 1) = \mathbb{P}(\tilde{Y} = 1 \mid X = x, A = 0) = Q_0(x),$$

so $\mathcal{U}(\tilde{Y}, x) = |Q_1(x) - Q_0(x)| = 0$ for all $x$.

However, the demographic disparity is (using the previous lemma that gives us the formula(18)):

$$\mathbb{P}(\tilde{Y} = 1 \mid A = 1) = \sum_{x \in \{0,1\}} \mathbb{P}(\tilde{Y} = 1 \mid X = x, A = 1)\mathbb{P}(X = x \mid A = 1)$$

$$= 0.2 \cdot \frac{1}{4} + 0.8 \cdot \frac{3}{4} = 0.05 + 0.6 = 0.65,$$

$$\mathbb{P}(\tilde{Y} = 1 \mid A = 0) = \sum_{x \in \{0,1\}} \mathbb{P}(\tilde{Y} = 1 \mid X = x, A = 0)\mathbb{P}(X = x \mid A = 0)$$

$$= 0.2 \cdot \frac{3}{4} + 0.8 \cdot \frac{1}{4} = 0.15 + 0.2 = 0.35.$$

Thus,

$$\Delta_{\mathrm{DP}}(\tilde{Y}) = |0.65 - 0.35| = 0.3 \neq 0.$$

$\square$

## D. Asymptotic Fairness Consistency and interpretable associated conditions

First let us start with the finite annotators sample bound:

**Proposition D.1.** *Let $R \in \mathbb{N}^\star$ denote the crowd size and $\phi \in \{\phi^\star, \phi^{\mathrm{MV}}\}$ be the aggregation rule. The demographic parity gap of the aggregated label $\hat{Y}^\phi$ converges to that of the ground-truth $Y$ with the following non-asymptotic bound:*

$$\left| \Delta_{\mathrm{DP}}(\hat{Y}_R^\phi) - \Delta_{\mathrm{DP}}(Y) \right| \leq \sum_{a \in \{0,1\}} \mathbb{E}_{X \mid A=a} \left[ e^{-R \cdot K_\phi(a,X)} \right], \tag{19}$$

*where $K_\phi$ is defined 3.1.*

*Proof.* Let us start with the definition, one has that:

$$\left| \Delta_{\mathrm{DP}}(\hat{Y}^{\mathrm{MV}}) - \Delta_{\mathrm{DP}}(Y) \right| = \left| \left| \int_{\mathcal{X}} \mathbb{P}(Y^{\mathrm{MV}} = 1 \mid X = x, A = 1)\, d\mathbb{P}(x \mid 1) - \int_{\mathcal{X}} \mathbb{P}(Y^{\mathrm{MV}} = 1 \mid X = x, A = 0)\, d\mathbb{P}(x \mid 0) \right| \right.$$

$$\left. - \left| \int_{\mathcal{X}} \mathbb{P}(Y = 1 \mid X = x, A = 1)\, d\mathbb{P}(x \mid 1) - \int_{\mathcal{X}} \mathbb{P}(Y = 1 \mid X = x, A = 0)\, d\mathbb{P}(x \mid 0) \right| \right|$$

Using

$$\forall x, y \in \mathbb{R} \quad \left| |x| - |y| \right| \leq |x - y|$$

and then the triangular inequality, one has that:

$$\left| \Delta_{\mathrm{DP}}(\hat{Y}^{\mathrm{MV}}) - \Delta_{\mathrm{DP}}(Y) \right| \leq \int_{\mathcal{X}} \left| \mathbb{P}(Y^{\mathrm{MV}} = 1 \mid X = x, A = 1) - \mathbb{P}(Y = 1 \mid X = x, A = 1) \right| d\mathbb{P}(x \mid 1)$$

$$+ \int_{\mathcal{X}} \left| \mathbb{P}(Y^{\mathrm{MV}} = 1 \mid X = x, A = 0) - \mathbb{P}(Y = 1 \mid X = x, A = 0) \right| d\mathbb{P}(x \mid 0)$$

Using the fact that

$$\left| \mathbb{P}(A) - \mathbb{P}(B) \right| \leq \mathbb{P}\left( A \setminus B \cup B \setminus A \right)$$

and

$$\mathbb{P}\left( Y^{MV} \neq Y \mid A = a, X = x \right) \leq \exp{-R J_R(x, a)}.$$

This leads to :

$$\left| \Delta_{\mathrm{DP}}(\hat{Y}^{\mathrm{MV}}) - \Delta_{\mathrm{DP}}(Y) \right| \leq \int_{\mathcal{X}} \exp{-R J_R(1, x)} d\mathbb{P}(x \mid 1) + \int_{\mathcal{X}} \exp{-R J_R(0, x)} d\mathbb{P}(x \mid 0).$$

This proves the proposition. $\square$

### D.1. Conditions for the Majority Vote:

Assume the annotators are divided into three disjoint sets:

$$\mathcal{G}_R = \{r : p_r \geq 1/2 + \varepsilon\}, \quad |\mathcal{G}_R| = g_{1,R},$$
$$\mathcal{B}_R = \{r : C \leq p_r \leq 1/2 - \varepsilon'\}, \quad |\mathcal{B}_R| = g_{2,R},$$
$$\mathcal{M}_R = (\mathcal{G}_R \cup \mathcal{B}_R)^c, \quad |\mathcal{M}_R| = R - g_{1,R} - g_{2,R},$$

where $C, \varepsilon, \varepsilon' > 0$.

**Lemma D.2.** *Let*

$$J_R(a, x) = - \min_{t \in (0,1]} \frac{1}{R} \sum_{r=1}^{R} \ln \left( p_r(a, x)t + \frac{1 - p_r(a, x)}{t} \right).$$

*Then, for any such partition,*

$$J_R(a, x) \geq - \min_{t \in (0,1]} \left[ \frac{g_{1,R}}{R} \ln \left( \left( \frac{1}{2} + \varepsilon \right) t + \frac{\frac{1}{2} - \varepsilon}{t} \right) + \frac{g_{2,R}}{R} \ln \left( Ct + \frac{1 - C}{t} \right) + \frac{R - g_{1,R} - g_{2,R}}{R} \ln \left( \frac{1}{t} \right) \right].$$

*Proof.* For a fixed $t \in (0, 1]$, define the function

$$f(p, t) = \ln \left( pt + \frac{1 - p}{t} \right).$$

We first show that $f(p, t)$ is decreasing in $p$ for fixed $t \in (0, 1]$.

$$\frac{\partial f}{\partial p}(p, t) = \frac{t - \frac{1}{t}}{pt + \frac{1-p}{t}}.$$

Since $t \in (0, 1]$, we have $t \leq 1$, so $t - \frac{1}{t} \leq 0$. it follows that,

$$\frac{\partial f}{\partial p}(p, t) \leq 0,$$

so $p \mapsto f(p, t)$ is decreasing for any given $t \in ]0, 1]$.

We split the sum in 3:

$$\frac{1}{R} \sum_{r=1}^{R} f(p_r, t) = \frac{1}{R} \left( \sum_{r \in \mathcal{G}_R} f(p_r, t) + \sum_{r \in \mathcal{B}_R} f(p_r, t) + \sum_{r \in \mathcal{M}_R} f(p_r, t) \right).$$

We bound each sum separately using the monotonicity of $p \mapsto f(p, t)$:

1. For $r \in \mathcal{G}_R$, we have $p_r \geq 1/2 + \varepsilon$. Since $f(., t)$ is decreasing , one has that:

$$f(p_r, t) \leq f(1/2 + \varepsilon, t) \quad \forall r \in \mathcal{G}_R.$$

   Therefore,

$$\sum_{r \in \mathcal{G}_R} f(p_r, t) \leq g_{1,R} \cdot f(1/2 + \varepsilon, t).$$

2. For $r \in \mathcal{B}_R$, we have $C \leq p_r \leq 1/2 - \varepsilon'$. Again,

$$f(p_r, t) \leq f(C, t) \quad \forall r \in \mathcal{B}_R,$$

   so one has that:

$$\sum_{r \in \mathcal{B}_R} f(p_r, t) \leq g_{2,R} \cdot f(C, t).$$

3. For $r \in \mathcal{M}_R$, $p_r \in [0, 1] \setminus (\mathcal{G}_R \cup \mathcal{B}_R)$. The worst-case scenario for maximizing $f(p_r, t)$ occurs at the smallest possible $p_r$, which is $0$. Hence,

$$f(p_r, t) \leq f(0, t) = - \ln t.$$

   one has that,

$$\sum_{r \in \mathcal{M}_R} f(p_r, t) \leq -(R - g_{1,R} - g_{2,R}) \ln t.$$

Combining these three bounds, we obtain for any $t \in (0, 1]$:

$$\frac{1}{R} \sum_{r=1}^{R} f(p_r, t) \leq \frac{g_{1,R}}{R} f(1/2 + \varepsilon, t) + \frac{g_{2,R}}{R} f(A, t) - \frac{R - g_{1,R} - g_{2,R}}{R} \ln t.$$

Since this inequality holds for every $t \in (0, 1]$, one has that :

$$\inf_{t \in (0,1]} \frac{1}{R} \sum_{r=1}^{R} f(p_r, t) \leq \inf_{t \in (0,1]} \left[ \frac{g_{1,R}}{R} f(1/2 + \varepsilon, t) - \frac{g_{2,R}}{R} f(C, t) - \frac{R - g_{1,R} - g_{2,R}}{R} \ln t \right].$$

$$J_R(a, x) \geq - \inf_{t \in (0,1]} \left[ \frac{g_{1,R}}{R} \ln \left( \left( \frac{1}{2} + \varepsilon \right) t + \frac{\frac{1}{2} - \varepsilon}{t} \right) + \frac{g_{2,R}}{R} \ln \left( At + \frac{1 - A}{t} \right) - \frac{R - g_{1,R} - g_{2,R}}{R} \ln t \right].$$

$\square$

**Proposition D.3.** *Let $\varepsilon > 0$, $C \in (0, \frac{1}{2})$ be fixed, and assume $0 < \varepsilon < \frac{1}{2}$ so that $\frac{1}{2} - \varepsilon > 0$. Define the fractions*

$$\alpha_R := \frac{g_{1,R}}{R}, \qquad \beta_R := \frac{g_{2,R}}{R}, \qquad \gamma_R := \frac{1 - \alpha_R - \beta_R}{R}.$$

*Consider the function*

$$F_R(t) = \alpha_R \ln \left( (\tfrac{1}{2} + \varepsilon) t + \frac{\frac{1}{2} - \varepsilon}{t} \right) + \beta_R \ln \left( Ct + \frac{1 - C}{t} \right) - \gamma_R \ln t, \qquad t \in (0, 1].$$

*If*

$$\liminf_{R \to \infty} \left( \alpha_R (1 + 2\varepsilon) + 2C\beta_R \right) > 1,$$

*then*

$$R \inf_{t \in (0,1]} F_R(t) \xrightarrow[R \to \infty]{} -\infty.$$

**Lemma D.4** (Intermediate Lemma). *Let $(f_n)_{n \in \mathbb{N}} \in \left( C^1(]0, 1], \mathbb{R}) \right)^{\mathbb{N}}$ satisfy:*

1. $\forall n \in \mathbb{N} \quad f_n(1) = 0$,

2. $\exists \delta > 0 \quad \forall n \in \mathbb{N} \quad f_n'(1) > \delta$,

3. $\exists \epsilon > 0 \quad \forall n \in \mathbb{N} \quad f_n'(t) > \dfrac{\delta}{2} \quad$ *for all $t \in [1 - \epsilon, 1]$.*

*Then*

$$\exists c > 0, \quad \exists N_0 \in \mathbb{N} \quad \forall n \geq N_0 \quad \inf_{r \in ]0,1]} f_n(r) \leq -c.$$

*Proof.* By condition (3), for each $n$ we have $f_n'(t) > \delta/2$ on $[1 - \epsilon, 1]$. Apply the mean value theorem to $f_n$ on $[1 - \epsilon, 1]$: there exists $\theta_n \in (1 - \epsilon, 1)$ such that

$$f_n(1 - \epsilon) = f_n(1) - \epsilon f_n'(\theta_n) = -\epsilon f_n'(\theta_n) < -\epsilon \cdot \frac{\delta}{2} = -\frac{\delta \epsilon}{2}.$$

Hence,

$$\inf_{r \in (0,1]} f_n(r) \leq f_n(1 - \epsilon) \leq -\frac{\delta \epsilon}{2}.$$

Taking $c = \delta \epsilon / 2 > 0$ completes the proof (the inequality holds for all $n$, so $N_0 = 1$ suffices). $\square$

We introduce the notion of equicontinuity of a familly of function $(f_n)_n \subset C([a, b], \mathbb{R})^{\mathbb{N}}$:

**Definition D.5.** $(f_n)_n$ is said to be equicontinue if and only if:

$$\forall x \in [a, b] \quad \forall \varepsilon > 0 \quad \exists \delta > 0 \quad \forall y \in [a, b] \quad \forall n \in \mathbb{N} : \quad |x - y| < \delta \implies |f_n(x) - f_n(y)| \leq \varepsilon. \tag{20}$$

**Lemma D.6.** *Assume that* $(f_n)_n \subset C([a, b], \mathbb{R})^{\mathbb{N}}$ *is equicontinue on* $[a, b]$ *and assume that*

$$\exists \omega > 0 \quad \forall n \quad f_n(b) > \omega, \tag{21}$$

*then one can find* $\varepsilon > 0$ *such that*

$$\forall n \in \mathbb{N} \quad \inf_{t \in [b-\varepsilon, b]} f_n(t) \geq \frac{\omega}{2}.$$

*Proof.* Set $x = b$ and $\varepsilon > 0$ in 20:

$$\exists \delta > 0 \quad \forall y \in [a, b] \quad \forall n \in \mathbb{N} : \quad |b - y| < \delta \implies |f_n(b) - f_n(y)| \leq \varepsilon.$$

$$\forall n \quad \forall y \in [b - \delta, b] \quad -\varepsilon + f_n(b) \leq f_n(y) \underset{21}{\Longrightarrow} \forall n \quad \forall y \in [b - \delta, b] \quad -\varepsilon + \omega \leq f_n(y).$$

Choose $\varepsilon = \frac{\omega}{2} > 0$ and the proof is finished.

$\square$

**Lemma D.7.** *Assume that* $(f_n)_n \subset C^1([a, b], \mathbb{R})^{\mathbb{N}}$ *, if*

$$\exists M > 0 \quad \forall n \quad \sup_{t \in [a, b]} |f_n'(t)| \leq M, \tag{22}$$

*then* $(f_n)$ *is equicontinous on* $[a, b]$

*Proof.* The proof is straightforward using the mean value theorem. $\square$

Here is the proof of D.3.

*Proof.* The structure of the proof is to verify the 3 conditions of D.4.

**Condition** 1: Since $\ln(1) = 0$ we have, $\forall R \geq 1$, $F_R(1) = 0$.

**Condition** 2: First, we compute the derivative of $F_R$:

$$F_R'(t) = \alpha_R \frac{(\frac{1}{2} + \varepsilon) - (\frac{1}{2} - \varepsilon)t^{-2}}{(\frac{1}{2} + \varepsilon)t + (\frac{1}{2} - \varepsilon)t^{-1}} + \beta_R \frac{C - (1 - C)t^{-2}}{Ct + (1 - C)t^{-1}} - \gamma_R t^{-1}.$$

At $t = 1$, this simplifies to

$$F_R'(1) = \alpha_R \frac{(\frac{1}{2} + \varepsilon) - (\frac{1}{2} - \varepsilon)}{(\frac{1}{2} + \varepsilon) + (\frac{1}{2} - \varepsilon)} + \beta_R \frac{C - (1 - C)}{C + (1 - C)} - \gamma_R = \alpha_R(1 + 2\varepsilon) + 2\beta_R C - 1.$$

Since by assumption

$$\liminf_{R \to \infty} (\alpha_R(1 + 2\varepsilon) + 2C\beta_R) > 1,$$

then there exists $\delta > 0$ and $R_0$ such that for all $R \geq R_0$,

$$\alpha_R(1 + 2\varepsilon) + 2C\beta_R \geq 1 + \delta.$$

Consequently, $F_R'(1) \geq \delta > 0$ for all $R \geq R_0$.

**Condition** 3: Here we use D.6 with $(f_R)_{R \in \mathbb{N}} = (F'_R)_{R \in \mathbb{N}}$. To apply it we need to show that $(F'_R)_{\mathbb{N}}$ is equicontinous and thanks to D.7 we just need to show that

$$\exists M > 0 \quad \forall R \quad \sup_{t \in [\frac{1}{2}, 1]} |F''_R(t)| \leq M. \tag{23}$$

Introduce :

$$\forall t \in ]0, 1] \quad U(t) = \frac{(\frac{1}{2} + \varepsilon)t^2 - (\frac{1}{2} - \varepsilon)}{(\frac{1}{2} + \varepsilon)t^2 + (\frac{1}{2} - \varepsilon)}, \qquad V(t) = \frac{Ct^2 - (1 - C)}{Ct^2 + (1 - C)}.$$

Then

$$\forall t \in ]0, 1] \quad F'_R(t) = \frac{1}{t} \big[ \alpha_R U(t) + \beta_R V(t) - \gamma_R \big].$$

$$\forall t \in ]0, 1] \quad F''_R(t) = -\frac{1}{t^2} \big[ \alpha_R U(t) + \beta_R V(t) - \gamma_R \big] + \frac{1}{t} \big[ \alpha_R U'(t) + \beta_R V'(t) \big].$$

For $t \in [1/2, 1]$: using the fact that neither $U, U'$ nor $V, V'$ does not depend on $R$ and are continuous function over the compact set $[\frac{1}{2}, 1]$, one can find

$$M_1 := \max \left( \sup_{t \in [\frac{1}{2}, 1]} |U(t)|, \sup_{t \in [\frac{1}{2}, 1]} |U'(t)|, \sup_{t \in [\frac{1}{2}, 1]} |V(t)|, \sup_{t \in [\frac{1}{2}, 1]} |V'(t)| \right) < \infty$$

$$\begin{aligned}
\sup_{t \in [\frac{1}{2}, 1]} |F''_R(t)| &\leq \sup_{t \in [\frac{1}{2}, 1]} |\alpha_R U(t) + \beta_R V(t) - \gamma_R| + \sup_{t \in [\frac{1}{2}, 1]} |\alpha_R U'(t) + \beta_R V'(t)| \\
&\leq M_1 (\alpha_R + \beta_R + \gamma_R + \alpha_R + \beta_R) \\
&\leq 2M_1.
\end{aligned}$$

Thus, there exists a constant $M := 2M_1 > 0$ such that

$$\forall t \in ]0, 1] \quad \forall R \geq 1 \quad |F''_R(t)| \leq M.$$

It follows that the family $(F'_R)_R$ is equicontinuous, on $[1/2, 1]$. Thus the condition (3) is verified.

*Remark* D.8. One can choose any $1 > a > 0$ instead of $\frac{1}{2}$ when considering the interval $[\frac{1}{2}, 1]$.

Applying lemma D.4 to $(F_R)_R$ it follows that one can find $c > 0$ (independante of $R$) such that for large $R$,

$$\inf_{t \in (0, 1]} F_R(t) \leq -c < 0,$$

which implies

$$R \inf_{t \in (0, 1]} F_R(t) \leq -cR \xrightarrow[R \to \infty]{} -\infty.$$

Conversely let us prove it by contraposition, if

$$\liminf_{R \to \infty} \big( \alpha_R(1 + 2\varepsilon) + 2C\beta_R \big) \leq 1,$$

then for a subsequence we have $F'_R(1) \leq 0$. Since $F_R(t) \to +\infty$ as $t \to 0^+$, the infimum is attained at some point $t_R \in (0, 1]$ and if $F'_R(1) \leq 0$, the infimum cannot be strictly less than $F_R(1) = 0$ (otherwise there would be a local minimum with positive derivative at 1). Thus $\inf F_R(t) = 0$ along that subsequence, and $R \inf F_R(t) = 0$ does not diverge to $-\infty$. $\qquad \square$

## D.2. Condition for the Bayesian Vote

**Lemma D.9.** *Assume that the annotators are not asymptotically random, in the sense that:*

$$\sum_{r=1}^{\infty} \left( p_r(X,a) - \frac{1}{2} \right)^2 = \infty \quad \mathbb{P}_X\text{-almost surely.} \tag{24}$$

*Then, the Bayesian error exponent guarantees convergence:*

$$\lim_{R\to\infty} e^{-RK_{\phi^\star}(x,a)} = 0 \quad \text{almost surely.}$$

*Proof.* Recall that the error exponent is given by $K_{\phi^\star}(x,a) = -\frac{1}{R}\sum_{r=1}^{R} \ln g(p_r(X,a))$, where $g(p) = 2\sqrt{p(1-p)}$. Let $S_R = -RK_{\phi^\star}(x,a) = \sum_{r=1}^{R} \ln g(p_r)$.

First, observe that $g(p) \leq 1$ for all $p \in [0,1]$, with equality if and only if $p = 1/2$. Consequently, $\ln g(p) \leq 0$, with $\ln g(1/2) = 0$. Consider the function $h : [0,1] \to \mathbb{R}$ defined by:

$$h(p) = \begin{cases} \frac{-\ln g(p)}{(p-1/2)^2} & \text{if } p \neq 1/2, \\ 2 & \text{if } p = 1/2. \end{cases}$$

Using a Taylor expansion of $\ln g(p)$ around $p = 1/2$, one can verify that $\lim_{p\to 1/2} h(p) = 2$, ensuring that $h$ is continuous on the compact interval $[0,1]$. Since $h(p) > 0$ for all $p$, it attains a global minimum $u := \min_{p \in [0,1]} h(p) > 0$.

This implies :

$$-\ln g(p) \geq u \left( p - \frac{1}{2} \right)^2 \quad \text{for all } p \in [0,1]. \implies -\sum_{r=1}^{R} \ln g(p_r) \geq u \sum_{r=1}^{R} \left( p_r - \frac{1}{2} \right)^2.$$

By the assumption (9), it implies that:

$$S_R = \sum_{r=1}^{R} \ln g(p_r) \xrightarrow{R\to\infty} -\infty \quad \text{a.s.}$$

$\square$

## D.3. Proof of Fairness consitency under interpretable conditions

*Proof of Theorem 3.4.* We apply the convergence dominated theorem to get the limite into the $\int$. Under assumptions and previous lemmas we get:

$$\forall (x,a) \in \{0,1\} \times \mathcal{X} \quad \lim_{R\to\infty} \exp{-RK_\phi(x,a)} = 0.$$

Applying dominated convergence theorem for each $a$ with measure $d\mathbb{P}(x \mid a)$ with the bounding function (in $R$) defined as

$$g_a : x \mapsto 1 \in L^1 \left( d\mathbb{P}(x \mid a) \right)$$

plus

$$\sup_{x \in X} \exp{-RK_\phi(x,a)} \leq 1$$

one has that

$$\lim_{R\to\infty} \Delta_{\text{DP}}(Y^{MV}) = \Delta_{\text{DP}}(Y).$$

$\square$

## E. Epsilon fairness of Majority Vote

**Theorem E.1** ((Baillon et al., 2015)). *Let $S_n = X_1 + ... + X_n$ be a sum of independent Bernoulli trials with probabilities* $\mathbb{P}(X_i = 1) = p_i$. *Let $\sigma_n^2 = \sum_{i=1}^{n} p_i(1 - p_i)$ denote the variance of the sum. There exists a universal constant $\eta$ such that:*

$$\forall n \geq 1, \quad \forall k \in \{0, ..., n\}, \quad \mathbb{P}(S_n = k) \leq \frac{\eta}{\sigma_n}. \tag{25}$$

*The optimal constant is given by:*

$$\eta = \max_{\lambda \geq 0} \sqrt{2\lambda} e^{-2\lambda} \sum_{k=0}^{\infty} \left(\frac{\lambda^k}{k!}\right)^2 \approx 0.4688. \tag{26}$$

**Lemma E.2.** *Let $Y_1, ..., Y_R$ be independent Bernoulli random variables with parameters $l_1, ..., l_R$. Define*

$$F(l_1, ..., l_R) = \mathbb{P}\left(\sum_{r=1}^{R} Y_r > \frac{R}{2}\right).$$

*Then, for every $r \in [R]$, we have*

$$\frac{\partial F}{\partial l_r} = \mathbb{P}\left(\sum_{s \neq r} Y_s = \lfloor R/2 \rfloor\right).$$

*Proof.* The cas when $R$ is odd is the same.

Let $R = 2m + 1$. Then $\lfloor R/2 \rfloor = m$, for any $r \in [1, R]$ one has that:

$$\left\{\sum_{l=1}^{R} Y_l \geq m + 1\right\} = \left\{\sum_{l \neq r}^{R} Y_l + Y_r \geq m + 1\right\}$$

$$= \left\{\sum_{l \neq r}^{R} Y_l \geq m \cap Y_r = 1\right\} \cup \left\{\sum_{l \neq r}^{R} Y_l \geq m + 1 \cap Y_r = 0\right\}$$

This shows:

$$\forall r \quad F(l_1, ..., l_n) = l_r \cdot \mathbb{P}\left(\sum_{s \neq r} Y_s \geq m\right) + (1 - l_r) \cdot \mathbb{P}\left(\sum_{s \neq r} Y_s \geq m + 1\right).$$

Derivating with respect to $l_r$ one has that:

$$\frac{\partial F}{\partial l_r} = \mathbb{P}\left(\sum_{s \neq r} Y_s \geq m\right) - \mathbb{P}\left(\sum_{s \neq r} Y_s \geq m + 1\right) = \mathbb{P}\left(\sum_{s \neq r} Y_s = m\right).$$

$\square$

**Proposition E.3.** *Let $Y_1, ..., Y_R$ be random variables that are conditionally independent given $A$ and $X$. Let $\hat{Y}_R^{\phi^{\mathrm{MV}}}$ denote the aggregated label from the Majority Vote.*

$$\epsilon(R) := \eta \left(\min\left\{\sqrt{V_R(0)}, \sqrt{V_R(1)}\right\}.\right)^{-1}$$

*Then, for all $R \geq 2$,*

$$\Delta_{\mathrm{DP}}(\hat{Y}_R^{\phi^{\mathrm{MV}}}) \leq \epsilon(R) \sum_{r=1}^{R} \Delta_{\mathrm{DP}}(\tilde{Y}_r). \tag{27}$$

*Proof.* Let $a \in \{0, 1\}$, first notice that:

$$\mathbb{P}(\hat{Y}_R^{\phi^{\mathrm{MV}}} = 1 \mid A = a) = \mathbb{P}\left(\sum_{r=1}^{R} Y_r > R/2 \mid A = a\right) = F(l(a))$$

and

$$\mathbb{P}(\hat{Y}_R^{\phi^{\mathrm{MV}}} = 0 \mid A = a) = 1 - F(p(a)),$$

where $F : (l_1, ..., l_r) \mapsto \mathbb{P}\left(\sum_{r=1}^{R} Y_r > R/2\right)$ with $Y_r \sim B(l_r)$. Observing that the Mean Value Theorem is invariant under the transformation $f \mapsto 1 - f$, it suffices to establish an upper bound for $Y^{\mathrm{MV}} = 1 \mid A = a$ for all $a$.

One can find $\theta \in [0, 1]$ such that :

$$F(l(1)) - F(l(0)) = \sum_{r=1}^{R} \frac{\partial F}{\partial l_r}(p(\theta)) \cdot (l_r(1) - l_r(0)),$$

where $l(\theta) = \theta l(1) + (1 - \theta) l(0)$.

Thanks to 3.5:

$$\frac{\partial F}{\partial l_r}(l(\theta)) = \mathbb{P}_{l(\theta)}\left(\sum_{s \neq r} Y_s = \lfloor R/2 \rfloor\right).$$

But the random variable $\sum_{s \neq r} Y_s \mid A = a$ follows a Bernoulli $R - 1$ (since $s \neq r$) of probability the vector $l(a)$.

Thus, we have that:

$$\sup_{\theta \in [0,1]} |\frac{\partial F}{\partial l_r}(l(\theta))| \underbrace{\leq}_{E.1} \sup_{\theta \in [0,1]} \frac{\eta}{\sqrt{\displaystyle\sum_{1 \leq i \leq R-1} (l(\theta))_i (1 - l(\theta)_i)}}$$

$$\implies \sup_{\theta \in [0,1]} |\frac{\partial F}{\partial l_r}(l(\theta))| \leq \frac{\eta}{\displaystyle\min_{\theta \in [0,1]} \sqrt{\displaystyle\sum_{1 \leq i \leq R-1} (l(\theta))_i (1 - l(\theta)_i)}}$$

$$\implies \sup_{\theta \in [0,1]} |\frac{\partial F}{\partial l_r}(l(\theta))| \leq \frac{\eta}{\min\left\{\sqrt{\sum_{i=1}^{R-1} l_i(0)(1 - l_i(0))}, \sqrt{\sum_{i=1}^{R-1} l_i(1)(1 - l_i(1))}\right\}}.$$

The proof is finished. □

# F. Proof of Theorem 4.1

Denote by $s_a = 2a - 1$ and $\pi_a = \mathbb{P}(A = a)$ for any $a \in \{0, 1\}$. In the following, in probabilities, we confound $\hat{Y} \sim \mathrm{Bernoulli}(g(W, A))$ and $g(W, A)$ such that the event $(g(W, A) = k)$ should be read $(\hat{Y} = k)$. Denote by $\mathcal{W}$ the input space without sensitive feature, which is $\mathcal{W} = \{0, 1\}^R \times \mathcal{X}$ in the crowdsourcing setting. Let $\mathcal{G}_\epsilon = \{g : \mathcal{W} \times [0, 1] \to \{0, 1\} \mid |\mathbb{P}(Y^g(W, A) = 1 | A = 1) - \mathbb{P}(Y^g(W, A) = 1 | A = 0)| \leq \epsilon, Y^g \sim \mathrm{Bernoulli}(g(W,A))\}$.

**Lemma F.1.** *The problem*

$$\arg\min_{\phi \in \mathcal{G}_\epsilon} \mathcal{R}, \quad \mathcal{R}(\phi) = \mathbb{P}(\phi(W, A) \neq Y)$$

*has the following Lagragian:*

$$\mathcal{R}_{\lambda^{(1)}, \lambda^{(2)}}(g) = \sum_{k=0}^{1} \sum_{a=0}^{1} \mathbb{E}_{W|A=a}(\mathbb{I}_{g(W,A) \neq k}\left(\pi_a P_k^*(W, a) - s_a s_k(\frac{\lambda^{(1)} - \lambda^{(2)}}{2})\right)) - \epsilon(\lambda^{(1)} + \lambda^{(2)}) \tag{28}$$

*Proof.* First, we consider the following Lagrangian for any random classifier $g$,

$$\mathcal{R}_{\lambda^{(1)},\lambda^{(2)}}(g) = \mathbb{P}(g(W,A) \neq Y) + \lambda^{(1)}[\mathbb{P}(g(W,A)=1|A=1) - \mathbb{P}(g(W,A)=1|A=0) - \epsilon] \tag{29}$$

$$+ \lambda^{(2)}[\mathbb{P}(g(W,A)=1|A=0) - \mathbb{P}(g(W,A)=1|A=1) - \epsilon] \tag{30}$$

The Lagrange multiplier $\lambda^{(1)}$ is related to the inequality $\mathbb{P}(g(W,A)=1|A=1) - \mathbb{P}(g(W,A)=1|A=0) \leq \epsilon$ while $\lambda^{(2)}$ is related to $\mathbb{P}(g(W,A)=1|A=0) - \mathbb{P}(g(W,A)=1|A=1) \leq \epsilon$. First, note that $\mathcal{R}_{\lambda^{(1)},\lambda^{(2)}}(g) = $

$$= \mathbb{P}(g(W,A) \neq Y) + \mathbb{P}(g(W,A)=1|A=1)(\lambda^{(1)} - \lambda^{(2)}) - \mathbb{P}(g(W,A)=1|A=0)(\lambda^{(1)} - \lambda^{(2)}) \tag{31}$$

$$- \epsilon(\lambda^{(1)} + \lambda^{(2)}) \tag{32}$$

$$= \mathbb{P}(g(W,A) \neq Y) + (1 - \mathbb{P}(g(W,A) \neq 1|A=1))(\lambda^{(1)} - \lambda^{(2)}) - (1 - \mathbb{P}(g(W,A) \neq 1|A=0))(\lambda^{(1)} - \lambda^{(2)}) \tag{33}$$

$$- \epsilon(\lambda^{(1)} + \lambda^{(2)}) \tag{34}$$

$$= \mathbb{P}(g(W,A) \neq Y) - \sum_{a=0}^{1} s_a(\lambda^{(1)} - \lambda^{(2)})\mathbb{E}_{W|A=a}[\mathbb{I}_{g(W,A)\neq 1}] - \epsilon(\lambda^{(1)} + \lambda^{(2)}) \tag{35}$$

$$= \mathbb{P}(g(W,A) \neq Y) - \sum_{k=0}^{1}\sum_{a=0}^{1} s_a s_k\left(\frac{\lambda^{(1)} - \lambda^{(2)}}{2}\right)\mathbb{E}_{W|A=a}[\mathbb{I}_{g(W,A)\neq k}] - \epsilon(\lambda^{(1)} + \lambda^{(2)}) \tag{36}$$

where in the last line we use that $x = 2x/2$ and $\mathbb{E}_{W|A=a}[\mathbb{I}_{g(W,A)\neq 1}] = 1 - \mathbb{E}_{W|A=a}[\mathbb{I}_{g(W,A)\neq 0}]$, the last constant 1 cancels because of the factor $s_a$. Applying similar computations yields

$$\mathbb{P}(g(W,A) \neq Y) = \sum_{k=0}^{1} \mathbb{E}(\mathbb{I}_{g(W,A)\neq k}\mathbb{I}_{Y=k}) = \sum_{k=0}^{1}\sum_{a=0}^{1} \mathbb{E}(\mathbb{I}_{g(W,A)\neq k}\mathbb{I}_{A=a}P_k^*(W,A)) \tag{37}$$

$$= \sum_{k=0}^{1}\sum_{a=0}^{1} \mathbb{E}_{W|A=a}(\mathbb{I}_{g(W,A)\neq k}\pi_a P_k^*(W,a)) \tag{38}$$

Finally,

$$\mathcal{R}_{\lambda^{(1)},\lambda^{(2)}}(g) = \sum_{k=0}^{1}\sum_{a=0}^{1} \mathbb{E}_{W|A=a}\left(\mathbb{I}_{g(W,A)\neq k}\left(\pi_a P_k^*(W,a) - s_a s_k\left(\frac{\lambda^{(1)} - \lambda^{(2)}}{2}\right)\right)\right) - \epsilon(\lambda^{(1)} + \lambda^{(2)}) \tag{39}$$

$\square$

**Proposition F.2.** *Let*

$$g^*_{\lambda^{(1)},\lambda^{(2)}} = \operatorname{argmin}_g \mathcal{R}_{\lambda^{(1)},\lambda^{(2)}}(g), \tag{40}$$

*then*

$$g^*_{\lambda^{(1)},\lambda^{(2)}}(w,a) = \begin{cases} 1, & \text{if } 2\pi_a P_1^*(w,a) > \pi_a + s_a(\lambda^{(1)} - \lambda^{(2)}), \\ 0, & \text{if } 2\pi_a P_1^*(w,a) < \pi_a + s_a(\lambda^{(1)} - \lambda^{(2)}), \\ \tau_{w,a}, & \text{if } 2\pi_a P_1^*(w,a) = \pi_a + s_a(\lambda^{(1)} - \lambda^{(2)}), \end{cases}$$

*with* $\tau_{w,a} \in [0,1]$.

*Proof.* Using Lemma F.1 and noting that the term in $\varepsilon$ does not depend on $g$, it suffices to minimize

$$\sum_{k=0}^{1}\sum_{a=0}^{1} \mathbb{E}_{W|A=a}\left[\mathbb{I}_{g(W,A)\neq k}\left(\pi_a P_k^*(W,a) - s_a s_k\frac{\lambda^{(1)} - \lambda^{(2)}}{2}\right)\right].$$

Using $P_0^*(W,a) = 1 - P_1^*(W,a)$, we rewrite the objective as

$$\sum_{a=0}^{1} \mathbb{E}_{W|A=a}\left[\mathbb{I}_{g(W,A)\neq 1}\left(\pi_a P_1^*(W,a) - s_a s_1\frac{\lambda^{(1)} - \lambda^{(2)}}{2}\right)\right.$$

$$\left. + \mathbb{I}_{g(W,A)\neq 0}\left(\pi_a(1 - P_1^*(W,a)) - s_a s_0\frac{\lambda^{(1)} - \lambda^{(2)}}{2}\right)\right].$$

Since the expectation is linear and the integrand depends on $g$ pointwise, we can minimize it for each fixed $(w, a)$ independently. For $k \in \{0, 1\}$, note that $s_1 = 1$ and $s_0 = -1$. For each $(w, a)$, define

$$A(w, a) = \pi_a P_1^*(w, a) - s_a \cdot 1 \cdot \frac{\lambda^{(1)} - \lambda^{(2)}}{2}, \quad B(w, a) = \pi_a(1 - P_1^*(w, a)) - s_a \cdot (-1) \cdot \frac{\lambda^{(1)} - \lambda^{(2)}}{2}.$$

The contribution for $(w, a)$ is

$$C(g; w, a) = \mathbb{I}_{g(w,a) \neq 1} A(w, a) + \mathbb{I}_{g(w,a) \neq 0} B(w, a).$$

We choose $g(w, a) \in \{0, 1\}$ to minimize this $C(g; w, a)$.

- If $g(w, a) = 1$, then $C(1; w, a) = B(w, a)$.
- If $g(w, a) = 0$, then $C(0; w, a) = A(w, a)$.
- If $g(w, a) = \tau \in [0, 1]$ then $\mathbb{E}[C(\tau; w, a)] = (1 - \tau)A(w, a) + \tau B(w, a)$.

So it follows that the minimizer can only be:

- $g(w, a) = 1$ if $B(w, a) < A(w, a)$,
- $g(w, a) = 0$ if $B(w, a) > A(w, a)$,
- Any $\tau \in [0, 1]$ if $B(w, a) = A(w, a)$.

$$
\begin{aligned}
B(w, a) - A(w, a) &= \left[\pi_a(1 - P_1^*(w, a)) + s_a \frac{\lambda^{(1)} - \lambda^{(2)}}{2}\right] - \left[\pi_a P_1^*(w, a) - s_a \frac{\lambda^{(1)} - \lambda^{(2)}}{2}\right] \\
&= \pi_a(1 - 2P_1^*(w, a)) + s_a(\lambda^{(1)} - \lambda^{(2)}).
\end{aligned}
$$

Hence,

$$B(w, a) < A(w, a) \iff \pi_a(1 - 2P_1^*(w, a)) + s_a(\lambda^{(1)} - \lambda^{(2)}) < 0 \iff 2\pi_a P_1^*(w, a) > \pi_a + s_a(\lambda^{(1)} - \lambda^{(2)}).$$

Similarly,

$$B(w, a) > A(w, a) \iff 2\pi_a P_1^*(w, a) < \pi_a + s_a(\lambda^{(1)} - \lambda^{(2)}),$$

and equality gives the indifferent case. $\qquad \square$

**Lemma F.3** (Reduction of the dual problem). *Under the same assumptions as in Proposition 1, define the dual objective*

$$\mathcal{H}(\lambda^{(1)}, \lambda^{(2)}) = \mathcal{R}_{\lambda^{(1)}, \lambda^{(2)}}(g_{\lambda^{(1)}, \lambda^{(2)}}^*),$$

*where $\mathcal{R}_{\lambda^{(1)}, \lambda^{(2)}}$ is the regularized risk and $g_{\lambda^{(1)}, \lambda^{(2)}}^*$ is the optimal classifier given by Proposition 1. Then:*

1. *The dual objective can be written as*

$$\mathcal{H}(\lambda^{(1)}, \lambda^{(2)}) = 1 - \mathcal{M}(\lambda^{(1)}, \lambda^{(2)}),$$

   *where*

$$
\begin{aligned}
\mathcal{M}(\lambda^{(1)}, \lambda^{(2)}) = \sum_{k=0}^1 \sum_{a=0}^1 \mathbb{E}_{W \mid A=a} &\Big[\mathbb{I}_{\{g_{\lambda^{(1)}, \lambda^{(2)}}^*(W, A) = k\}} \\
&\times \Big(\pi_a P_k^*(W, a) - s_a s_k \frac{\lambda^{(1)} - \lambda^{(2)}}{2}\Big)\Big] + \epsilon(\lambda^{(1)} + \lambda^{(2)}).
\end{aligned}
$$

   *Consequently, maximizing $\mathcal{H}$ over $\lambda^{(1)}, \lambda^{(2)} \geq 0$ is equivalent to minimizing $\mathcal{M}$.*

2. *There exists a function $\mathcal{L} : \mathbb{R} \to \mathbb{R}$ such that*

$$\mathcal{M}(\lambda^{(1)}, \lambda^{(2)}) = \mathcal{L}(\lambda^{(1)} - \lambda^{(2)}) + \epsilon(\lambda^{(1)} + \lambda^{(2)}),$$

   *where*

$$\mathcal{L}(\beta) = \sum_{a=0}^1 \mathbb{E}_{W \mid A=a}\left[\max_{k \in \{0,1\}}\Big(\pi_a P_k^*(W, a) - s_a s_k \frac{\beta}{2}\Big)\right].$$

3. *At any optimal solution $(\lambda_*^{(1)}, \lambda_*^{(2)})$ of the dual problem, we have $\lambda_*^{(1)} \cdot \lambda_*^{(2)} = 0$. Hence, letting $\beta = \lambda_*^{(1)} - \lambda_*^{(2)}$, it holds that $\lambda_*^{(1)} + \lambda_*^{(2)} = |\beta|$ and therefore*

$$\mathcal{M}(\lambda_*^{(1)}, \lambda_*^{(2)}) = \mathcal{L}(\beta) + \epsilon|\beta|.$$

*Proof.*    1. From the definition of the risk $\mathcal{R}_{\lambda^{(1)}, \lambda^{(2)}}$ (see Lemma F.1) and the optimal classifier $g^*_{\lambda^{(1)}, \lambda^{(2)}}$, we have

$$\mathcal{H}(\lambda^{(1)}, \lambda^{(2)}) = \sum_{k=0}^{1} \sum_{a=0}^{1} \mathbb{E}_{W \mid A=a} \Big[ \big(1 - \mathbb{I}_{\{g^*_{\lambda^{(1)}, \lambda^{(2)}}(W,A)=k\}}\big)$$

$$\times \Big( \pi_a P_k^*(W, a) - s_a s_k \frac{\lambda^{(1)} - \lambda^{(2)}}{2} \Big) \Big] - \epsilon(\lambda^{(1)} + \lambda^{(2)}).$$

Observe that

$$\sum_{k=0}^{1} \sum_{a=0}^{1} \mathbb{E}_{W \mid A=a} \big[ \pi_a P_k^*(W, a) \big] = \sum_{a=0}^{1} \pi_a \sum_{k=0}^{1} \mathbb{P}(Y = k \mid A = a) = \sum_{a=0}^{1} \pi_a = 1.$$

2. The dependence on $\lambda^{(1)}, \lambda^{(2)}$ only through $\beta = \lambda^{(1)} - \lambda^{(2)}$ in the first term follows from the form of the optimal classifier $g^*_{\lambda^{(1)}, \lambda^{(2)}}$, whose decision rule (Proposition 1) depends on $\pi_a + s_a(\lambda^{(1)} - \lambda^{(2)})$. So, the indicator $\mathbb{I}_{\{g^*=k\}}$ is a function of $\beta$ only. Define

$$\mathcal{L}(\beta) = \sum_{k=0}^{1} \sum_{a=0}^{1} \mathbb{E}_{W \mid A=a} \Big[ \mathbb{I}_{\{g^*_\beta(W,A)=k\}} \Big( \pi_a P_k^*(W, a) - s_a s_k \frac{\beta}{2} \Big) \Big],$$

where $g^*_\beta$ denotes the optimal classifier with $\lambda^{(1)} - \lambda^{(2)} = \beta$. To show the max representation, fix $(w, a)$ and consider the two possible choices of $k$. From the derivation of the optimal classifier, $g^*_\beta(w, a)$ chooses the label $k$ to maximizing $\pi_a P_k^*(w, a) - s_a s_k \beta / 2$. Therefore,

$$\sum_{k=0}^{1} \mathbb{I}_{\{g^*_\beta(w,a)=k\}} \Big( \pi_a P_k^*(w, a) - s_a s_k \frac{\beta}{2} \Big) = \max_{k \in \{0,1\}} \Big( \pi_a P_k^*(w, a) - s_a s_k \frac{\beta}{2} \Big),$$

even in the case of equality where randomization is allowed. Taking expectations gives you $\mathcal{L}(\beta)$.

3. Suppose $(\lambda^{(1)}, \lambda^{(2)})$ is a minimizer of $\mathcal{M}$ with both $\lambda^{(1)} > 0$ and $\lambda^{(2)} > 0$. For any $\delta > 0$ small enough, consider $(\lambda^{(1)} - \delta, \lambda^{(2)} - \delta)$. Then $\beta = \lambda^{(1)} - \lambda^{(2)}$ remains unchanged, but $\lambda^{(1)} + \lambda^{(2)}$ decreases by $2\delta$. Since $\mathcal{M}(\lambda^{(1)}, \lambda^{(2)}) = \mathcal{L}(\beta) + \epsilon(\lambda^{(1)} + \lambda^{(2)})$, we have

$$\mathcal{M}(\lambda^{(1)} - \delta, \lambda^{(2)} - \delta) = \mathcal{L}(\beta) + \epsilon(\lambda^{(1)} + \lambda^{(2)} - 2\delta) < \mathcal{M}(\lambda^{(1)}, \lambda^{(2)}),$$

contradicting minimality. Hence, at least one of $\lambda^{(1)}$ or $\lambda^{(2)}$ must be zero. If $\beta \geq 0$, then $\lambda^{(1)} = \beta$ and $\lambda^{(2)} = 0$; if $\beta < 0$, then $\lambda^{(1)} = 0$ and $\lambda^{(2)} = -\beta$. In both cases, $\lambda^{(1)} + \lambda^{(2)} = |\beta|$, leading to $\mathcal{M} = \mathcal{L}(\beta) + \epsilon|\beta|$.

$\square$

**Proposition F.4** (Existence of a minimizer). *Define $\mathcal{M}(\beta) = \mathcal{L}(\beta) + \epsilon|\beta|$ with $\mathcal{L}$ as in Lemma F.3 and $\epsilon > 0$. Then:*

1. *The function $\mathcal{M}$ is convex.*

2. *It is coercive: $\lim_{|\beta| \to \infty} \mathcal{M}(\beta) = +\infty$.*

3. *So, there exists at least one global minimizer $\beta^* \in \mathbb{R}$ of $\mathcal{M}$.*

*Proof.*    1. For each fixed $(w, a)$, the function

$$\beta \mapsto \max_{k \in \{0,1\}} \Big( \pi_a P_k^*(w, a) - s_a s_k \frac{\beta}{2} \Big)$$

is the pointwise maximum of two affine functions, hence convex. Taking expectation over $W$ given $A = a$ preserves convexity. The absolute value function $\beta \mapsto |\beta|$ is convex, and the sum of two convex functions is convex. Therefore, $\mathcal{M}$ is convex.

2. To show coercivity, examine the behavior of $\mathcal{L}(\beta)$ as $|\beta| \to \infty$. Recall that for $s_a = 1$:

$$\max_k \left( \pi_a P_k^*(w, a) - s_a s_k \frac{\beta}{2} \right) = \max \left\{ \pi_a P_0^*(w, a) + \frac{\beta}{2}, \ \pi_a P_1^*(w, a) - \frac{\beta}{2} \right\}.$$

For $s_a = -1$:

$$\max_k \left( \pi_a P_k^*(w, a) - s_a s_k \frac{\beta}{2} \right) = \max \left\{ \pi_a P_0^*(w, a) - \frac{\beta}{2}, \ \pi_a P_1^*(w, a) + \frac{\beta}{2} \right\}.$$

There exists a constant $C > 0$, depending only on $\pi_a, P_0^\star, P_1^\star$ such that for all $|\beta|$ sufficiently large,

$$\mathcal{L}(\beta) \geq \frac{1}{2} |\beta| - C(\pi_a, P_0^\star, P_1^\star).$$

Hence,

$$\mathcal{M}(\beta) = \mathcal{L}(\beta) + \epsilon |\beta| \geq \left( \frac{1}{2} + \epsilon \right) |\beta| - C(\pi_a, P_0^\star, P_1^\star) \xrightarrow[|\beta| \to \infty]{} +\infty.$$

3. A convex function that is coercive attains its minimum on $\mathbb{R}$. Thus, there exists $\beta^* \in \mathbb{R}$ such that $\mathcal{M}(\beta^*) = \inf_{\beta \in \mathbb{R}} \mathcal{M}(\beta)$.

$\square$

**Lemma F.5** (Subgradient condition and fairness). *Let $\beta^*$ be a minimizer of $\mathcal{M}(\beta) = \mathcal{L}(\beta) + \epsilon |\beta|$, where $\mathcal{L}$ is defined as in Lemma F.3. Define the classifier $g_{\beta^*}^*$ via Proposition 1 with $\lambda^{(1)} - \lambda^{(2)} = \beta^*$ (and $\lambda^{(1)} \lambda^{(2)} = 0$). Then:*

1. *There exist $(\omega_a)_{a \in \{0,1\}} \in [0,1]^2$ such that $(\tau_a = \omega_a)_{a \in \{0,1\}}$ used on the event $(2\pi_a P_a^*(w, a) = \pi_a + s_a \beta^*)$ in Theorem F.2 are solution of*

$$\xi^* \epsilon = \mathbb{P}\big( g_{\beta^*}^*(W, A) = 1 \mid A = 1 \big) - \mathbb{P}\big( g_{\beta^*}^*(W, A) = 1 \mid A = 0 \big), \tag{41}$$

*where $\xi^* \in \partial_\beta |\cdot|\big|_{\beta = \beta^*}$ satisfies $\xi^* = \operatorname{sign}(\beta^*)$ if $\beta^* \neq 0$ and $\xi^* \in [-1, 1]$ if $\beta^* = 0$. The existence derives from the optimality condition $0 \in \partial \mathcal{M}(\beta^*)$ and the unicity holds as long as $\omega_a$ are chosen with minimal norm $|\omega_0| + |\omega_1|$.*

2. *Consequently, by setting $\tau_a = \omega_a$*

$$\left| \mathbb{P}\big( g_{\beta^*}^*(W, A) = 1 \mid A = 1 \big) - \mathbb{P}\big( g_{\beta^*}^*(W, A) = 1 \mid A = 0 \big) \right| = \begin{cases} \epsilon, & \text{if } \beta^* \neq 0, \\ \leq \epsilon, & \text{if } \beta^* = 0. \end{cases}$$

*That is, $g_{\beta^*}^*$ is $\epsilon$-fair.*

*Proof.* Let $f^a(w, \beta) = \max_{k \in \{0,1\}} \big( \pi_a P_k^*(w, a) - s_a s_k \beta/2 \big)$. For each $(w, a)$, the subdifferential $\partial_\beta f^a(w, \beta)$ is given by

$$\partial_\beta f^a(w, \beta) = \begin{cases} \{-s_a/2\}, & \text{if } 2\pi_a P_1^*(w, a) > \pi_a + s_a \beta, \\ \{s_a/2\}, & \text{if } 2\pi_a P_1^*(w, a) < \pi_a + s_a \beta, \\ [-1/2, 1/2], & \text{if } 2\pi_a P_1^*(w, a) = \pi_a + s_a \beta. \end{cases}$$

Since $\mathcal{L}(\beta) = \sum_{a=0}^1 \mathbb{E}_{W \mid A = a}[f^a(W, \beta)]$, by the subgradient interchange property, we have

$$\partial \mathcal{L}(\beta) = \sum_{a=0}^1 \mathbb{E}_{W \mid A = a} \big[ \partial_\beta f^a(W, \beta) \big].$$

Let $F_a(\beta) = \mathbb{P}\big( 2\pi_a P_1^*(W, a) > \pi_a + s_a \beta \mid A = a \big)$ and $\bar{F}_a(\beta) = \mathbb{P}\big( 2\pi_a P_1^*(W, a) = \pi_a + s_a \beta \mid A = a \big)$. Then, for any $\beta$,

$$\mathbb{E}_{W \mid A = a} \big[ \partial_\beta f^a(W, \beta) \big] = -\frac{s_a}{2} F_a(\beta) + \frac{s_a}{2} \big( 1 - F_a(\beta) - \bar{F}_a(\beta) \big) + \left[ -\tfrac{1}{2}, \tfrac{1}{2} \right] \bar{F}_a(\beta),$$

where the interval $[-\frac{1}{2}, \frac{1}{2}]$ is for the subgradient on the equality set. Simplifying,

$$\mathbb{E}_{W\,|\,A=a}\big[\partial_\beta f^a(W, \beta)\big] = -\frac{s_a}{2}\big(2F_a(\beta) + \bar{F}_a(\beta) - 1\big) + \big[-\tfrac{1}{2}, \tfrac{1}{2}\big]\bar{F}_a(\beta).$$

Hence, there exist $\mu_a \in [-1/2, 1/2]$ such that

$$\partial\mathcal{L}(\beta) \ni \sum_{a=0}^{1}\left[-\frac{s_a}{2}\big(2F_a(\beta) + \bar{F}_a(\beta) - 1\big) + \mu_a\bar{F}_a(\beta)\right].$$

The optimality condition $0 \in \partial\mathcal{M}(\beta^*)$ gives $0 \in \partial\mathcal{L}(\beta^*) + \epsilon\partial|\beta^*|$. Let $\xi^* \in \partial|\beta^*|$ with $\xi^* = \mathrm{sign}(\beta^*)$ if $\beta^* \neq 0$ and $\xi^* \in [-1, 1]$ if $\beta^* = 0$. Then,

$$0 = \sum_{a=0}^{1}\left[-\frac{s_a}{2}\big(2F_a(\beta^*) + \bar{F}_a(\beta^*) - 1\big) + \mu_a\bar{F}_a(\beta^*)\right] + \epsilon\xi^*.$$

Rearranging,

$$\epsilon\xi^* = \sum_{a=0}^{1}\frac{s_a}{2}\big(2F_a(\beta^*) + \bar{F}_a(\beta^*) - 1\big) - \sum_{a=0}^{1}\mu_a\bar{F}_a(\beta^*).$$

Moreover, by setting $\omega_a = 1/2 - \mu_a \in [0, 1]$, we have by the definition of $g^*_{\beta*}$ we have that

$$\mathbb{P}\big(g^*_{\beta*}(W, a) = 1\,|\,A = a\big) = F_a(\beta^*) + \omega_a\bar{F}_a(\beta^*),$$

. Consequently,

$$\begin{aligned}
\epsilon\xi^* &= \sum_{a=0}^{1} s_a\big(F_a(\beta^*) + \tfrac{1}{2}\bar{F}_a(\beta^*)\big) - \sum_{a=0}^{1}\frac{s_a}{2} - \sum_{a=0}^{1}\mu_a\bar{F}_a(\beta^*) \\
&= \sum_{a=0}^{1} s_a\big(F_a(\beta^*) + \omega_a\bar{F}_a(\beta^*)\big) \\
&= \sum_{a=0}^{1} s_a\,\mathbb{P}\big(g^*_{\beta*}(W, a) = 1\,|\,A = a\big).
\end{aligned}$$

It follows that:

$$\epsilon\xi^* = \mathbb{P}\big(g^*_{\beta*}(W, A) = 1\,|\,A = 1\big) - \mathbb{P}\big(g^*_{\beta*}(W, A) = 1\,|\,A = 0\big). \tag{42}$$

This proves (1). Statement (2) follows immediately because $|\xi^*| \leq 1$ and $\xi^* = \pm 1$ when $\beta^* \neq 0$.

Denoting by $\omega^F = \omega_1\bar{F}_1(\beta^*) - \omega_0\bar{F}_0(\beta^*) \in [-\bar{F}_0(\beta^*), \bar{F}_1(\beta^*)]$, $\omega_0, \omega^1$ are uniquely defined by $\omega^F$ if taken with minimal norm $|\omega_0| + |\omega_1|$ and $\omega^F$ is uniquely defined in (42) if chosen with minimal norm $\omega^F$,

$$\epsilon\xi^* = F_1(\beta^*) - F_0(\beta^*) + \omega^F,$$

this observation yields the unicity of $(\omega_a)$ when chosen with minimal norm.

$\square$

Here is the final step of the proof of theorem 4.1.

**Proposition F.6.** *Let* $\mathcal{G}_\epsilon = \{g : \mathcal{W} \times [0, 1] \to \{0, 1\}\,|\,|\mathbb{P}(Y^g(W, A) = 1|A = 1) - \mathbb{P}(Y^g(W, A) = 1|A = 0)| \leq \epsilon, Y^g \sim Bernoulli(g(W,A))\}$ *be the set of* $\epsilon$-*fair classifiers. Then the classifier* $g^*_{\beta*}$ *obtained from the dual solution (as in Lemma F.5) satisfies*

$$g^*_{\beta*} \in \mathrm{argmin}_{g\in\mathcal{G}_\epsilon}\,\mathbb{P}\big(g(W, A) \neq Y\big).$$

*That is,* $g^*_{\beta*}$ *minimizes the misclassification error among all* $\epsilon$-*fair classifiers.*

*Proof.* First, by Lemma F.5, $g^*_{\beta^*} \in \mathcal{G}_\epsilon$. Let $\lambda^{(1)}_* = \max(\beta^*, 0)$ and $\lambda^{(2)}_* = \max(-\beta^*, 0)$, so that $\lambda^{(1)}_* \lambda^{(2)}_* = 0$ and $\beta^* = \lambda^{(1)}_* - \lambda^{(2)}_*$. Recall the regularized risk

$$\mathcal{R}_{\lambda^{(1)}, \lambda^{(2)}}(g) = \mathbb{P}(g(W, A) \neq Y) + \lambda^{(1)} \big( \Delta_{\mathrm{DP}}(g) - \epsilon \big)_+ + \lambda^{(2)} \big( -\Delta_{\mathrm{DP}}(g) - \epsilon \big)_+,$$

where $\Delta_{\mathrm{DP}}(g) = \mathbb{P}(g(W, A) = 1 | A = 1) - \mathbb{P}(g(W, A) = 1 | A = 0)$ and $(x)_+ = \max(x, 0)$.

Since $g^*_{\beta^*}$ is $\epsilon$-fair, we have $|\Delta_{\mathrm{DP}}(g^*_{\beta^*})| \leq \epsilon$. Moreover, by Lemma F.5, if $\beta^* \neq 0$ then $|\Delta_{\mathrm{DP}}(g^*_{\beta^*})| = \epsilon$ and the sign of $\beta^*$ matches the sign of $\Delta_{\mathrm{DP}}(g^*_{\beta^*})$. Consequently,

$$\lambda^{(1)}_* \big( \Delta_{\mathrm{DP}}(g^*_{\beta^*}) - \epsilon \big)_+ + \lambda^{(2)}_* \big( -\Delta_{\mathrm{DP}}(g^*_{\beta^*}) - \epsilon \big)_+ = 0.$$

Therefore,

$$\mathcal{R}_{\lambda^{(1)}_*, \lambda^{(2)}_*}(g^*_{\beta^*}) = \mathbb{P}\big( g^*_{\beta^*}(W, A) \neq Y \big).$$

Now, for any other classifier $g \in \mathcal{G}_\epsilon$, we have $|\Delta_{\mathrm{DP}}(g)| \leq \epsilon$, so

$$\big( \Delta_{\mathrm{DP}}(g) - \epsilon \big)_+ = 0 \quad \text{and} \quad \big( -\Delta_{\mathrm{DP}}(g) - \epsilon \big)_+ = 0.$$

Hence,

$$\mathcal{R}_{\lambda^{(1)}_*, \lambda^{(2)}_*}(g) = \mathbb{P}\big( g(W, A) \neq Y \big).$$

But by definition, $g^*_{\beta^*}$ minimizes $\mathcal{R}_{\lambda^{(1)}_*, \lambda^{(2)}_*}$ (since it is the optimal classifier for those Lagrange multipliers). Thus,

$$\mathbb{P}\big( g^*_{\beta^*}(W, A) \neq Y \big) = \mathcal{R}_{\lambda^{(1)}_*, \lambda^{(2)}_*}(g^*_{\beta^*}) \leq \mathcal{R}_{\lambda^{(1)}_*, \lambda^{(2)}_*}(g) = \mathbb{P}\big( g(W, A) \neq Y \big).$$

This holds for every $g \in \mathcal{G}_\epsilon$, proving the claim. $\qquad\square$

We have prooved the theorem 4.1.

## G. Proof of Theorem A.1

The beginning of the proof mirrors the beginning of Theorem 4.1's proof.

In the following, in probabilities, we confound $\hat{Y} \sim \mathrm{Multiniouilli}(g(X, W))$ and $g(X, W)$ such that the event $(g(X, W) = k)$ should be read $(\hat{Y} = k)$.

First, we consider the following Lagrangian for any random classifier $g$,

$$\mathcal{R}_{\lambda^{(1)}, \lambda^{(2)}}(g) = \mathbb{P}(g(W, A) \neq Y) + \sum_{k=1}^{K} \lambda^{(1)}_k [\mathbb{P}(g(W, A) = 1 | A = 1) - \mathbb{P}(g(W, A) = 1 | A = 0) - \epsilon] \qquad (43)$$

$$+ \lambda^{(2)}_k [\mathbb{P}(g(W, A) = 1 | A = 0) - \mathbb{P}(g(W, A) = 1 | A = 1) - \epsilon] \qquad (44)$$

The Lagrange multiplier $\lambda^{(1)}_k$ is related to the inequality $\mathbb{P}(g(W, A) = k | A = 1) - \mathbb{P}(g(W, A) = k | A = 0) \leq \epsilon$ while $\lambda^{(2)}_k$ is related to $\mathbb{P}(g(W, A) = k | A = 0) - \mathbb{P}(g(W, A) = 1 | A = 1) \leq \epsilon$.

We aim to solve $\max_{\lambda^{(1)}, \lambda^{(2)} \in (\mathbb{R}^K_{\geq 0})^2} \min_{g \in \mathcal{G}} \mathcal{R}_{\lambda^{(1)}, \lambda^{(2)}}(g)$ and to show that the solution solves (11), where $\mathcal{G}$ is the set of random classifiers of interest.

First, note that $\mathcal{R}_{\lambda^{(1)}, \lambda^{(2)}}(g) =$

$$= \mathbb{P}(g(W, A) \neq Y) - \sum_{k=1}^{K} \left[ \sum_{a=0}^{1} s_a(\lambda^{(1)}_k - \lambda^{(2)}_k) \mathbb{E}_{W|A=a}\big( \mathbb{I}_{g(W,A) \neq k} \big) \right] - \epsilon(\lambda^{(1)}_k + \lambda^{(2)}_k) \qquad (45)$$

Applying similar computations yields

$$\mathbb{P}(g(W,A) \neq Y) = \sum_{k=1}^{K} \mathbb{E}(\mathbb{I}_{g(W,A)\neq k}\mathbb{I}_{Y=k}) = \sum_{k=1}^{K}\sum_{a=0}^{1} \mathbb{E}(\mathbb{I}_{g(W,A)\neq k}\mathbb{I}_{A=a}P_k^*(W,A)) \tag{46}$$

$$= \sum_{k=1}^{K}\sum_{a=0}^{1} \mathbb{E}_{W|A=a}(\mathbb{I}_{g(W,A)\neq k}\pi_a P_k^*(W,a)) \tag{47}$$

Finally,

$$\mathcal{R}_{\lambda^{(1)},\lambda^{(2)}}(g) = \sum_{k=1}^{K}\left[\sum_{a=0}^{1}\mathbb{E}_{W|A=a}(\mathbb{I}_{g(W,A)\neq k}\left(\pi_a P_k^*(W,a) - s_a(\lambda_k^{(1)} - \lambda_k^{(2)}))\right)\right] - \epsilon(\lambda_k^{(1)} + \lambda_k^{(2)}) \tag{48}$$

Defining

$$g_{\lambda^{(1)},\lambda^{(2)}}^* = \operatorname{argmin}_g \mathcal{R}_{\lambda^{(1)},\lambda^{(2)}}(g) \tag{49}$$

and minimizing the terms inside the expectations $\mathbb{E}_{W|a}$, denoting by $\beta_j = \lambda_j^{(1)} - \lambda_j^{(2)}$, we have

$$\phi_{\lambda^{(1)},\lambda^{(2)}}^*(w,a) = \begin{cases} (\mathbb{I}_{\{k\}}(i))_{i\in\{1,...,K\}}, & \text{if } \{k\} = \arg\max_{k\in\{1,...,K\}}\left(\pi_a P_j^*(w,a) - s_a\beta_j\right) \quad j\neq k \\ (\tau_{i,a}^{\mathsf{J}}\mathbb{I}_{\mathsf{J}}(i))_{i\in\{1,...,K\}}, & \text{if } \arg\max_{k\in\{1,...,K\}}\left(\pi_a P_k^*(w,a) - s_a\beta_k\right) = \mathsf{J}, \quad \mathsf{J}\subset\{1,...K\}, |\mathsf{J}|>1 \end{cases}$$

with $\tau_{i,a}^{\mathsf{J}} \in [0,1]$ and the only constraint on these values is $\sum_{i=1}^{K}\tau_{i,a}^{\mathsf{J}} = 1$.

Then, the risk on the dual variable is maximized

$$\lambda_*^{(1)}, \lambda_*^{(2)} = \operatorname{argmax}_{\lambda^{(1)},\lambda^{(2)}\in(\mathbb{R}_{\geq 0}^K)^2}\mathcal{H}(\lambda^{(1)},\lambda^{(2)}), \quad \mathcal{H}(\lambda^{(1)},\lambda^{(2)}) = \mathcal{R}_{\lambda^{(1)},\lambda^{(2)}}(g_{\lambda^{(1)},\lambda^{(2)}}^*) \tag{50}$$

Note that

$$\mathcal{H}(\lambda^{(1)},\lambda^{(2)}) = \sum_{k=1}^{K}\sum_{a=0}^{1}\mathbb{E}_{X|A=a}((1-\mathbb{I}_{g_{\lambda^{(1)},\lambda^{(2)}}^*(W,A)=k})\left(\pi_a P_k^*(W,a) - s_a(\lambda_k^{(1)} - \lambda_k^{(2)})\right) - \epsilon(\lambda_k^{(1)} + \lambda_k^{(2)}) \tag{51}$$

$$= \underbrace{\sum_{k=1}^{K}\sum_{a=0}^{1}\mathbb{E}_{W|A=a}(\pi_a P_k^*(W,a))}_{=1} \tag{52}$$

$$-\underbrace{\sum_{k=1}^{K}\sum_{a=0}^{1}\mathbb{E}_{W|A=a}(\mathbb{I}_{g_{\lambda^{(1)},\lambda^{(2)}}^*(W,A)=k}\left(\pi_a P_k^*(W,a) - s_a(\lambda_k^{(1)} - \lambda_k^{(2)}))\right) - \epsilon(\lambda_k^{(1)} + \lambda_k^{(2)})}_{\mathcal{M}(\lambda^{(1)},\lambda^{(2)})} \tag{53}$$

Therefore, maximizing $\mathcal{H}$ reduces to minimizing $\lambda^{(1)},\lambda^{(2)} \in (\mathbb{R}_{\geq 0}^K)^2 \mapsto \mathcal{M}(\lambda^{(1)},\lambda^{(2)})$. Since $\mathcal{M}(\lambda^{(1)},\lambda^{(2)}) = \mathcal{L}(\lambda^{(1)} - \lambda^{(2)}) + \epsilon(\lambda^{(1)} + \lambda^{(2)})$ where $\mathcal{L}: \mathbb{R} \mapsto \mathbb{R}$, by minimality principle, we have $\lambda^{(1)}\lambda^{(2)} = 0$ such that we replace the variables by $\beta_k = \lambda_k^{(1)}\mathbb{I}_{\lambda_k^{(1)}\geq 0} - \lambda_k^{(2)}\mathbb{I}_{\lambda_k^{(1)}>0}$, and $\mathcal{M}(\lambda^{(1)},\lambda^{(2)}) = \mathcal{L}(\beta) + \epsilon|\beta|$ with

$$\mathcal{L}(\beta) = \sum_{k=1}^{K}\sum_{a=0}^{1}\mathbb{E}_{W|A=a}(\mathbb{I}_{g_{\lambda^{(1)},\lambda^{(2)}}^*(W,A)=k}\left(\pi_a P_k^*(W,a) - s_a\beta_k)\right) \tag{54}$$

$$= \sum_{a=0}^{1}\mathbb{E}_{W|A=a}(\sum_{k=1}^{K}\mathbb{I}_{g_{\lambda^{(1)},\lambda^{(2)}}^*(W,A)=k}\left(\pi_a P_k^*(W,a) - s_a\beta_k)\right) \tag{55}$$

$$= \sum_{a=0}^{1}\mathbb{E}_{W|A=a}\left(\max_{k\in\{0,1\}}\pi_a P_k^*(W,a) - s_a\beta_k\right) \tag{56}$$

Thus, $\mathcal{M} : \beta \mapsto \mathcal{L}(\beta) + \epsilon|\beta|$ is convex as a sum of maximum of linear functions and $\lim_{\beta \to \infty} |\mathcal{M}(\beta)| = +\infty$, which implies the existence of a minimum $\beta^*$.

At a minimum $\beta^* \in \mathbb{R}^K$, there exists $0_K \in \partial\mathcal{M}(\beta^*)$ by optimality. In the following, $\partial\mathcal{M}(\beta^*)$ is described in detail to obtain a condition on the demographic parity of the classifier $g^*$, and on the $\tau_{i,a}^{\mathsf{J}}$.

Denote by $h_k^a : (w, \beta) \mapsto \pi_a P_k^*(w, a) - s_a \beta_k$ and $f^a : (w, \beta) \mapsto \max_k \pi_a P_k^*(w, a) - s_a \beta_k$, we have

$$\partial_\beta f^a = \mathrm{conv}\{\nabla_\beta h_k^a(w, \beta) \quad : \quad h_k^a(w, \beta) = \max_j h_j^a(w, \beta)\} \tag{57}$$

$$= \mathrm{conv}\{(-s_a \mathbb{I}_{\{k\}}(j))_{j \in \{1,\dots,K\}} \quad : \quad h_k^a(w, \beta) = \max_j h_j^a(w, \beta)\} \tag{58}$$

where $\mathrm{conv}\{(z_i)_{i \in \{1,\dots,M\}}\} = \{\sum_{i=1}^M u_i z_i : \sum_{i=1}^M u_i = 1, \, u_i \geq 0\}$.

For any $w$, $\chi \in \partial_\beta f^a(w, \beta)$, we have

$$\chi = \sum_{k=1}^K u_{k,a}(w) \nabla_\beta h_k^a(w, \beta) = (-s_a u_{k,a}(w))_{k \in \{1,\dots,K\}} \tag{59}$$

with $\sum_{k=1}^K u_{k,a}(w) = 1$, $u_{k,a}(w) \geq 1$ and $u_{k,a}(w) = 0$ when $h_k^a(w, \beta) < \max_j h_j^a(w, \beta)$.

Using the fact that $\partial(f + g) = \partial f + \partial g$ if $f$ or $g$ is continuous, then we have since $f^a$ is continuous,

$$\partial\mathbb{E}_{W|A=a}(f^a(W, \beta)) = \sum_{\mathsf{J} \subset \{1,\dots,K\}} \partial\mathbb{E}_{W|A=a}(\mathbb{I}_{\arg\max_j h_j^a(W,\beta)=\mathsf{J}} f^a(W, \beta)) \tag{60}$$

For any $\mathsf{J} \subset \{1, \dots, K\}$, $\chi \in \partial\mathbb{E}_{W|A=a}(\mathbb{I}_{\arg\max_j h_j^a(W,\beta)=\mathsf{J}} f^a(W, \beta))$, there exist $(\omega_{i,a}^{\mathsf{J}})_{i \in \{0,\dots,K\}} \in \mathbb{R}_{\geq 0}^K$ such that $\omega_{i,a}^{\mathsf{J}} = 0$ if $i \notin \mathsf{J}$ and $\sum_{i=1}^K \omega_{i,a}^{\mathsf{J}} = 1$,

$$\chi = \mathbb{P}(\arg\max_j h_j^a(W, \beta) = \mathsf{J}|A = a) \times (-s_a \omega_{i,a}^{\mathsf{J}})_{i \in \{0,\dots,K\}} \tag{61}$$

Using $0_K \in \partial\mathcal{M}(\beta^*)$, we deduce that for any $k \in \{1, \dots, K\}$ there exist $(\omega_{k,a}^{\mathsf{J}})_{k,a,\mathsf{J}}$ and $\xi_k \in [-1, 1]$ such that $\xi_k^* = \mathbb{I}_{\beta_k^* > 0} - \mathbb{I}_{\beta_k^* < 0} + \xi_k \mathbb{I}_{\beta_k^* = 0} \in \partial_\beta |\cdot|$,

$$0 = -\sum_{a=0}^1 \left[ s_a \sum_{\mathsf{J} \subset \{1,\dots,K\}} \mathbb{P}(\arg\max_j h_j^a(W, \beta) = \mathsf{J}|A = a)\omega_{k,a}^{\mathsf{J}} \right] + \xi_k^* \epsilon \tag{62}$$

$$0 = \sum_{a=0}^1 -s_a \mathbb{P}(h_k^a(W, \beta^*) > h_j^a(W, \beta^*), j \neq k|A = a)+ \tag{63}$$

$$- s_a \mu_{a,k} \mathbb{P}((h_k^a(W, \beta^*) > h_j^a(W, \beta^*), j \neq k) \cup (\exists i \neq k : h_k^a(W, \beta^*) = h_i^a(W, \beta^*))|A = a) + \xi_k^* \epsilon \tag{64}$$

where in the second line we use that $\mathbb{P}(h_k^a(W, \beta^*) > h_j^a(W, \beta^*), j \neq k|A = a) = \mathbb{P}(\arg\max_k h_k^a(W, \beta^*) = \{k\})$

$$\mathcal{P}_{k,a} = \mathbb{P}((h_k^a(W, \beta) > h_j^a(W, \beta^*), j \neq k) \cup (\exists i \neq k : h_k^a(W, \beta^*) = h_i^a(W, \beta^*))|A = a) \tag{65}$$

$$= \sum_{\mathsf{J} \subset \{1,\dots,K\}, \, k \in \mathsf{J}, |\mathsf{J}| > 1} \mathbb{P}(\arg\max_k h_k^a(W, \beta^*) = \mathsf{J}) \tag{66}$$

and we denote by

$$\mu_{k,a} = \sum_{\mathsf{J} \subset \{1,\dots,K\}, \, k \in \mathsf{J}, |\mathsf{J}| > 1} \omega_{a,i}^{\mathsf{J}} \frac{\mathbb{P}(\arg\max_k h_k^a(W, \beta^*) = \mathsf{J})}{\mathcal{P}_{k,a}} \in [0, 1] \tag{67}$$

Thus, by setting $\tau_{i,a}^{\mathsf{J}} = \omega_{i,a}^{\mathsf{J}}$, we have

$$0 = \sum_{a=0}^1 -s_a \mathbb{P}(\phi_{\beta^*}^* = k|A = a) + \xi_k^* \epsilon \tag{68}$$

$$\mathbb{P}(\phi_{\beta^*}^* = k|A = 1) - \mathbb{P}(\phi_{\beta^*}^* = k|A = 0) = \xi_k^* \epsilon \tag{69}$$

The end of the proof mirrors the proof of Theorem 4.1.

Now, we aim to show that $\phi_{\beta^*}^* = \operatorname{argmin}_{g \in \mathcal{G}_\epsilon} \mathbb{P}(g(W, A) \neq Y)$. First, $\phi_{\beta^*}^* \in \mathcal{G}_\epsilon$ by definition and by (68). Second, reminding that $\lambda_{*,k}^{(1)} = \mathbb{I}_{\beta^* \geq 0}\beta^*$, $\lambda_{*,k}^{(2)} = \mathbb{I}_{\beta_k^* < 0}|\beta_k^*|$, we have $\mathcal{R}_{\lambda_*^{(1)}, \lambda_*^{(2)}}(\phi_{\lambda_*^{(1)}, \lambda_*^{(2)}}^*) =$

$$\mathbb{P}(\phi_{\lambda_*^{(1)}, \lambda_*^{(2)}}^*(W, A) \neq Y) + \sum_{k=1}^{K} |\beta_k^*|(|\mathbb{P}(\phi_{\beta^*}^* = k|A = 1) - \mathbb{P}(\phi_{\beta^*}^* = k|A = 0)| - \epsilon) = \mathbb{P}(\phi_{\lambda_*^{(1)}, \lambda_*^{(2)}}^*(W, A) \neq Y)$$

where the equation follows by $|\mathbb{P}(\phi_{\beta^*}^* = k|A = 1) - \mathbb{P}(\phi_{\beta^*}^* = k|A = 0)| = \epsilon$ as long as $\beta_k^* \neq 0$ by (68). More generally, for any $g \in \mathcal{G}_\epsilon$,

$$\mathcal{R}_{\lambda_*^{(1)}, \lambda_*^{(2)}}(g) \leq \mathbb{P}(g(W, A) \neq Y) + \left(\sum_{k=1}^{K} |\beta_k^*|\right)(\Delta_{\mathrm{DP}}(g(W, A)) - \epsilon) \leq \mathbb{P}(g(W, A) \neq Y)$$

and thus,
$$\mathbb{P}(\phi_{\lambda_*^{(1)}, \lambda_*^{(2)}}^*(W, A) \neq Y) = \mathcal{R}_{\lambda_*^{(1)}, \lambda_*^{(2)}}(\phi_{\lambda_*^{(1)}, \lambda_*^{(2)}}^*) \leq \mathcal{R}_{\lambda_*^{(1)}, \lambda_*^{(2)}}(g) \leq \mathbb{P}(g(W, A) \neq Y)$$

which concludes the proof.

# H. Robustness to Dataset Imbalance

We further evaluate the robustness of our algorithm on an imbalanced hate speech dataset (Kennedy et al., 2020). This setting is challenging due to both class imbalance and sensitive-group imbalance.

Here accuracy can be misleading; we therefore report the F1-score, which better captures the precision–recall trade-off in this setting.

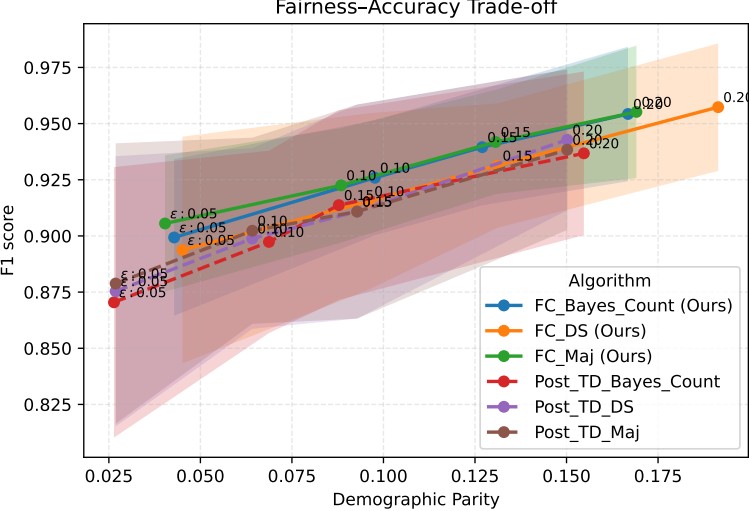

*Figure 5.* Performance comparison on the Measuring Hate Speech dataset with $A = 1$ if the comment targets a gender. Solid lines correspond to our method (FC), while dashed lines indicate competing approaches. Shaded regions denote the variance across 10 independent runs using different test sets (60%).

