# OpenReview forum: "Optimal Fair Aggregation of Crowdsourced Noisy Labels using Demographic Parity Constraints"
_ICML.cc/2026/Conference — ICML 2026 regular_

### Official Review · Reviewer_xYHm · 2026-03-08

**Soundness:** 3
**Presentation:** 2
**Significance:** 3
**Originality:** 3
**Overall Recommendation:** 4
**Confidence:** 1

**Summary:**

The paper studies the problem of fairly aggregating crowdsourced labels, i.e.,
we have multiple (noisy) crowdsourced labels for each record in a dataset,
and we want to aggregate them into a single label.
Each record has a sensitive attribute, and we don't want the distribution of labels
to vary a lot across the sensitive attribute; this is called fairness.
This paper quantifies fairness as the _Demographic Parity gap_ (DP gap).
The paper focuses on binary labels and a binary sensitive attribute,
though some of the techniques are generalizable to other types of labels.

There are two well-known methods of aggregating labels: Bayes-optimal and majority-vote.
For both, the paper proves that the DP gap converges exponentially fast to the
(unobservable) ground-truth label. It does so using the result by Gao et al
that the error probability converges exponentially fast to 0.
The paper also proves a few corollaries to make this result more interpretable.
Thus, the paper proves that the DP gap of these two aggregation methods
is not much worse than the DP gap of the ground truth.

However, sometimes the ground truth might itself be unfair.
The paper considers the problem of finding the aggregation rule
that minimizes the misclassification error
under the constraint that the DP gap should be at most $ε$.
They obtain a closed-form expression for the optimal solution,
and explain how to estimate this aggregator from data.
They call this the FairCrowd Algorithm, and it outperforms
other known methods on the fairness-accuracy frontier
when evaluated on real-world datasets.

**Compliance With Llm Reviewing Policy:**

Affirmed.

**Final Justification:**

The rebuttal addressed my main concerns. One of my concerns was quite serious (the paper seemed to be missing a proof), but since I had given the authors the benefit of doubt in my original review (i.e., I assumed that the proof was there but I couldn't find it), my score remains unchanged.

**Key Questions For Authors:**

1.  In equation (2), it's unclear what $y$ is. Did you mean
    $p_r(a, x) = P(\widetilde{Y}_r = Y | X=x, A=a)$?
2.  Where is the proof of Proposition 3.2?
    How do you relate the error probability to difference in DP gap?
3.  Page 5, line 240, column 2: What is $τ_a$?

**Limitations:**

Yes

**Strengths And Weaknesses:**

The paper studies an important problem.
I like the theoretical analysis of their methodology
and they improve upon the state-of-the-art in practice.

The paper could have been better written.
Section 3 (theoretical analysis) is mostly just a sequence of theorems,
with little context on their significance and how they relate to each other.

The paper seems to not include a proof of one of its claims (Proposition 3.2).
Maybe I couldn't find it in the paper. Hopefully the authors can clarify this.

---

> ### Author Rebuttal · Authors · 2026-03-31
>
> Thank you for your review and for your time, which will help us improve the clarity of the presentation and notations.
>
> **Question 1: Notation in Equation (2)**
> Yes, we understand your point. There is no dependency on $y$ in $p_r(a,x)$, because of the one coin model assumption we are making. We will clarify this point.
>
> **Question 2: Proof reference for Proposition 3.2**
>
> We thank the reviewer for spotting this. A clear pointer will be added to the proof of Proposition 3.2, which is already in Appendix D.
>
> **Question 3: Notation on page 5, line 240**
>
> $\tau_{a}$ is the probability with which the classifier randomizes (predicts $1$ or $0$) when the posterior probability falls exactly on the decision threshold of group $a$. To ensure consistency in our notations, we will change $\tau_{a}$ into $\omega_{a}$.

---

> > ### Author Rebuttal · Reviewer_xYHm · 2026-04-01
> >
> > I see now; you renamed Proposition 3.2 to Proposition D.1 in the appendix. That's why I couldn't find it. Use consistent numbering in the paper.

---

### Official Review · Reviewer_34nP · 2026-03-11

**Soundness:** 2
**Presentation:** 3
**Significance:** 3
**Originality:** 3
**Overall Recommendation:** 4
**Confidence:** 3

**Summary:**

This paper addresses the fairness issue in crowdsourcing annotation and proposes a unified theoretical and algorithmic framework. By simplifying the convergence conditions, this study derives, for the first time, the non-asymptotic convergence bounds of Majority Vote and Bayesian Aggregation under Demographic Parity. It is proven that, given qualified annotator quality, the fairness gap of aggregated labels converges to that of the true labels at an exponential rate.
Furthermore, to tackle the potential unfairness inherent in true labels, this work extends the $\epsilon$-fair post-processing framework for classifiers from continuous inputs to the discrete crowdsourcing scenario, and develops the FairCrowd algorithm, which can enforce fairness constraints for arbitrary aggregation rules.

**Compliance With Llm Reviewing Policy:**

Affirmed.

**Final Justification:**

Thank you for the detailed response. I will maintain my positive score 4.

**Key Questions For Authors:**

The conclusion states that "In particular, FairCrowd applied to Majority Vote is computationally efficient and performs well even without prior information." Could you clarify how the claimed computational efficiency and the robust performance in the absence of prior information should be interpreted?
How does the variance of algorithmic performance and fairness metrics behave under limited annotation redundancy, specifically when the average number of annotators per sample is low (2–3)?

**Limitations:**

Yes

**Strengths And Weaknesses:**

Strengths
The theory of this paper is rigorous. Through the analysis of the optimal inequality for the Poisson binomial distribution, it provides an upper bound on fairness in small-scale crowdsourcing.
The experimental part covers synthetic data and real benchmark datasets such as Crowd Judgement and Jigsaw Toxicity, which verifies the theoretical predictions and demonstrates the superiority of the algorithm in the trade-off between F1 score and fairness.
It fills the gap in the theoretical guarantee of fairness for crowdsourcing aggregation, especially when the true labels are unavailable and may contain biases, providing a fairness correction scheme that does not rely on the ground truth


Weaknesses
It should be noted that this paper only focuses on Demographic Parity as the fairness metric, while Equalized Odds and Equal Opportunity are more commonly used in many practical applications. However, this paper does not discuss whether the proposed method can be extended to these two fairness metrics.
Although the experiments adopt the F1-score as the evaluation metric, they do not demonstrate that the proposed method maintains robust F1 performance under highly class-imbalanced data distributions.
The implementation details are described somewhat briefly.
Figure 1 is overly simplistic, while the main text contains an excessive density of equations; a more balanced allocation of expository space between visual illustrations and mathematical formulations is recommended.
In contrast to "Towards Fair Truth Discovery from Biased Crowdsourced Answers", the present work does not report classification accuracy as an evaluation metric.

---

> ### Author Rebuttal · Authors · 2026-03-31
>
> We thank the reviewer for pressing us on the interpretation of our efficiency claims and robustness results. We have also improved Figure 1 according to your remark.
>
> Before answering the reviewer's questions, we come back to two remarks that helps us to improve the soundness of the paper.
>
> **Class imbalance and accuracy**
>
> The proportion of labeled samples is 14% in Jigsaw, 47% in Crowd Judgement, and 49% in Synthetic. In the following figure (https://imgur.com/a/YLw5jjY), we report the results on Jigsaw (which is notably imbalanced) using the macro F1-score, Accuracy, and F1-score binary. As shown, the results remain largely consistent with our previous findings. We did not choose to report Accuracy values, as they were aligned with the F1-scores, and would have been therefore redundant.
>
> In the revised manuscript, we plan to extend our benchmark in the appendix with additional real-world datasets from the industry that exhibit more diverse label distributions.
>
> **Considering Equalized odds**
>
> An extension to equalized odds is possible when full access to the ground-truth labels $Y$ is available: in that case, we can constrain the difference in positive prediction rates between groups for each true class. This results in a Lagrangian with $2 \times 2K$ constraints, two for each pair $(Y,a) \in {1,\ldots,K} \times {0,1}$. However, this would require controlling the decision boundary for every value of $(Y,a)$ during inference, which is not feasible in practice, unlike demographic parity that depends only on the sensitive attribute $(\pi_a \pm s_a \beta^*) / 2\pi_a$. A similar reasoning applies to equalized opportunity.
>
> We leave this limitation for future work, as it may be addressed by leveraging surrogate models for the ground-truth labels, and iterative procedures.
>
> **Computational efficiency and robustness to prior information**
>
> The concluding sentence on *computational efficiency* will be clarified. In this context, *computational efficiency* refers to the comparison between FairCrowd’s or PostTD's runtime (they operate typically in several seconds) and FairTD’s runtime (it needs at least one minute). It also highlights the simplicity of Majority Vote, which is essentially a straightforward averaging procedure, relative to other aggregation methods (DS, Bayes) that may require training steps and prior information.
>
> The phrase "robust without prior information" indicates that, despite not relying on learned priors, Majority Vote still achieves strong fairness-accuracy trade-offs on average across the datasets in our benchmark.
>
> **Variance under low annotation redundancy**
>
> With only a few annotators per sample (i.e. 2 to 3), the variance in fairness and accuracy is not necessarily higher than with dense annotations. This is illustrated in the Jigsaw experiment, where more than half of the dataset has fewer than 4 annotations, yet the variance remains relatively stable when using Majority Vote. In practice, performance variance is more sensitive to the number of samples in the training or test sets than to the number of annotators per sample.
>
> To further highlight this point, we will include additional experiments on synthetic data in the appendix.

---

> > ### Author Rebuttal · Reviewer_34nP · 2026-04-07
> >
> > Thank you for the detailed response. I will maintain my positive score 4.

---

### Official Review · Reviewer_QkGR · 2026-03-13

**Soundness:** 3
**Presentation:** 3
**Significance:** 3
**Originality:** 3
**Overall Recommendation:** 4
**Confidence:** 4

**Summary:**

This paper studies fairness in crowdsourced label aggregation, focusing on how biases among annotators affect the fairness properties of aggregated labels. In many ML settings where reliable ground-truth labels are unavailable or expensive to obtain, labels are collected from multiple annotators and aggregated using methods such as Majority Vote or Bayesian aggregation.

The paper investigates what fairness guarantees these aggregation procedures have under the demographic parity (DP) criterion. The authors derive non-asymptotic bounds on the demographic parity gap of aggregated labels and show that this gap converges exponentially fast to the fairness gap of the ground-truth label under certain conditions on annotator quality (Theorem 3.2). The analysis also considers the small-crowd regime and shows that Majority Vote can amplify bias when annotator biases are aligned (Theorem 3.6).

To mitigate fairness issues, the authors propose a post-processing algorithm called FairCrowd (FC), which enforces ε-fairness constraints under demographic parity. As I understand it, the approach extends an ε-fairness post-processing framework to the crowdsourcing setting and to discrete input spaces.

The paper evaluates the method on synthetic data and two crowdsourcing benchmarks (Crowd Judgement and Jigsaw toxicity). These experiments illustrate fairness–accuracy trade-offs and compare FairCrowd to existing approaches such as Post_TD and FairTD.

**Compliance With Llm Reviewing Policy:**

Affirmed.

**Final Justification:**

The rebuttal provides helpful clarifications regarding the scope of the method and its limitations, particularly around the use of demographic parity and the role of the aggregation model in shaping annotator bias. These responses address my main concerns and improve my understanding of the contribution. While the overall assessment of the paper remains unchanged, the rebuttal reinforces my view that the theoretical analysis is the primary strength of the work and that the contribution has practical utility.

**Key Questions For Authors:**

- Do the authors think the FairCrowd framework could be extended to other fairness notions such as equalized odds?

- How would the method behave if annotator biases depend on input features rather than only on sensitive attributes?

- Would similar fairness amplification effects appear when using more sophisticated aggregation models such as neural aggregation approaches?

**Limitations:**

The paper focuses only on demographic parity as the fairness notion. The theoretical analysis also assumes relatively simple annotator behavior and independence between annotators. In addition, the empirical evaluation covers only a small number of datasets.

**Strengths And Weaknesses:**

### Strengths

- I think the paper addresses an important but somewhat overlooked problem. There has been extensive work on fairness during model training, but fairness issues that arise during label aggregation itself have received much less attention.

- The theoretical analysis is, in my view, one of the strongest parts of the paper. The non-asymptotic bounds on the demographic parity gap for Majority Vote and Bayesian aggregation (Theorem 3.2) provide a useful lens for understanding how annotator quality and crowd size affect fairness of aggregated labels.

- I also found the bias amplification result in the small-crowd regime (Theorem 3.6) interesting. It formalizes the intuition that when annotator biases are aligned, Majority Vote can reinforce those biases rather than average them out.

- The FairCrowd algorithm is simple and fairly flexible. Since it operates as a post-processing step, it can be applied on top of different aggregation rules without requiring changes to the aggregation procedure itself.

- The experiments provide a useful illustration of fairness–accuracy trade-offs. For example, Figures 3 and 4 show how enforcing ε-fairness affects F1 score on the Crowd Judgement and Jigsaw datasets.

---

### Weaknesses

- One limitation, in my opinion, is that the paper focuses exclusively on demographic parity. While this choice makes sense in crowdsourcing settings where ground-truth labels may not be available, demographic parity has well-known limitations and may not align with other fairness notions such as equalized odds. I would have liked to see some discussion of whether the framework could extend to other fairness criteria.

- The algorithmic contribution seems somewhat incremental. The FairCrowd method mostly adapts the ε-fairness post-processing framework of Denis et al. (2024) to the crowdsourcing setting. While this extension is useful, my impression is that the main novelty of the paper lies more in the theoretical analysis than in the algorithm itself.

- The theoretical results rely on fairly simplified annotator assumptions, including independence between annotators and relatively simple error models. In practice, annotator errors can be correlated or depend on task features. I think it would strengthen the paper to discuss how sensitive the guarantees are to these assumptions.

- The empirical evaluation is somewhat limited. The experiments use synthetic data and two datasets (Crowd Judgement and Jigsaw toxicity). I would have liked to see either additional datasets or experiments with more diverse aggregation settings.

- One experimental detail that stood out is that Bayesian aggregation performs best on the synthetic and Crowd Judgement datasets but not on Jigsaw, where Majority Vote performs better. This behavior could use a bit more explanation in the paper.

---

> ### Author Rebuttal · Authors · 2026-03-31
>
> Thank you for your time and valuable comments.
>
> First, we acknowledge that including an additional dataset would have strengthened our comparison, and we expect to provide supplementary industrial examples in the appendix of the final manuscript. However, note that there is no theoretical reason to expect PostTD or FairTD—both of which do not rely on posterior estimates $\hat{\Phi}$—to outperform FairCrowd, which uses posterior estimates produced by any aggregation method.
>
>
>  Here are the responses to the questions you raised.
>
>
>
> **Question 1: extension to equalized odds**
>
>
> An extension to equalized odds is possible when full access to the ground‑truth labels $Y$ is available: in that case, we can constrain the difference in positive prediction rates between groups for each true class. This results in a Lagrangian with $2 \times 2K$ constraints, two for each pair $(Y,a) \in {1,\ldots,K} \times {0,1}$. However, this would require controlling the decision boundary for every value of $(Y,a)$ during inference, which is not feasible in practice—unlike demographic parity, which depends only on the sensitive attribute $(\pi_a \pm s_a \beta^*) / 2\pi_a$. A similar reasoning applies to equalized opportunity.
>
> We leave this limitation for future work, as it may be addressed by leveraging surrogate models for the ground‑truth labels, and iterative procedures.
>
> **Question 2: Feature-dependent annotator biases**
>
> In our experiments, we estimate the annotators’ confusion matrices under the assumption that
> $\pi^{(r)}_{y',y}(a)$ does not depend on the features.
> This limitation arises from the **aggregation method’s modeling choice** for the posterior estimates $\hat{\Phi}(w,a)$, rather than from our postprocessing method, which simply takes these posterior estimates as input and is therefore agnostic to such assumptions. Our framework does, in theory, allow feature‑dependent annotator skills (Theorem 3.1); in particular, $w$ may include $(\tilde{Y}, X)$ as discussed at the beginning of Section 4.
> Incorporating feature‑dependent annotator skill or bias would improve overall performance only if the posterior estimates $\hat{\Phi}(w,a)$ themselves become more accurate.
>
>
>
> In practice, however, reliably estimating feature‑conditional confusion matrices poses identifiability challenges when annotation redundancy is limited (e.g., the Jigsaw dataset has a median of only four labels per item), as also noted in the review of (Ibrahim et al. 2025). For this reason, we opt for a simpler approach in the experiments. Nonetheless, for completeness, we plan to include additional experiments in the appendix using recent crowdsourcing methods that incorporate both features and sensitive attributes.
>
> **Question 3: Sophisticated aggregation models (e.g., neural)**
>
>
> If we aim to generalize the amplification result (Proposition3.6), the sharp probability bounds rely on the Poisson-binomial framework (Baillon et al., 2015), which applies uniquely to unweighted majority vote. For weighted aggregation rules (e.g., Bayesian vote, neural networks), the decision function becomes:
> $\phi(\tilde{Y}, X, A) = \mathbf{1}\left({\sum\limits_{i=1}^{R} w_i(X, A) \tilde{Y}_i \geq \tau(X, A)}\right),$
> where weights $w_i$ and threshold $\tau$ are data-dependent. Point probability bounds for weighted Bernoulli sums remain an open combinatorial problem.
>
> Experimentally, for neural aggregator methods, we expect the amplification effect to be mitigated in a manner similar to the Dawid–Skene aggregation (Figure 2), since prior information is used to improve the estimation even with a small number of annotators $R$.
>
> We thank the reviewer for this comment, which will help improve the clarity of our paper and motivates us to further develop the extension to equalized odds.

---

> > ### Author Rebuttal · Reviewer_QkGR · 2026-04-02
> >
> > I appreciate the authors’ detailed and thoughtful responses. The discussion around extending to equalized odds was helpful, and I agree that the reliance on demographic parity is partly motivated by the lack of ground-truth labels in crowdsourcing settings. The response regarding feature-dependent annotator biases clarifies that this limitation mainly comes from the aggregation model rather than the post-processing step itself. On the question of more sophisticated aggregation methods, I think the answer is reasonable.
> >
> > Overall, the rebuttal improves my understanding of the scope and limitations of the work and adequately addresses my main concerns. I still view the main strength of the paper as the theoretical analysis, which provides a useful characterization of fairness in crowdsourced label aggregation, and believe the contribution has practical utility.

---

### Official Review · Reviewer_wESF · 2026-03-16

**Soundness:** 3
**Presentation:** 4
**Significance:** 3
**Originality:** 2
**Overall Recommendation:** 4
**Confidence:** 3

**Summary:**

This paper examines fairness in crowdsourced label aggregation under demographic-parity constraints. It provides theoretical analysis for Majority Vote and Bayesian aggregation, extends the $\epsilon$-fair post-processing framework to the crowdsourcing setting, and proposes the FairCrowd algorithm.

**Compliance With Llm Reviewing Policy:**

Affirmed.

**Final Justification:**

I support the acceptance of this paper.

**Key Questions For Authors:**

L1. The theory is still somewhat incomplete from a practical perspective. The strongest guarantees are asymptotic or small-crowd results, while finite-sample fairness guarantees are missing.

L2. The algorithm relies on sequential quadratic programming and grid search, but does not provide convergence or complexity analysis.

L3. One practical limitation should be stated more clearly: in experiments, the estimation procedure does not model interactions between sensitive attributes and inputs, which may reduce realism in settings with feature-dependent annotator bias.

**Limitations:**

See L1 - L3.

**Strengths And Weaknesses:**

S1. The topic is important and relatively underexplored: fairness at the label aggregation stage is a meaningful problem beyond standard downstream classifier fairness.

S2. The proposed framework is principled: FairCrowd is formulated through constrained post-processing instead of heuristic label flipping. Experiments on synthetic and real datasets support the main claim that fairness can be improved with a reasonable accuracy tradeoff.

---

> ### Author Rebuttal · Authors · 2026-03-31
>
> We thank the reviewer for his comments. Addressing them led to a stronger theoretical analysis of our framework. We now provide explicit non-asymptotic concentration bounds and rigorous convergence and complexity analysis.
>
> **L1:**
> To get a non-asymptotic bound, we derive explicit closed-form bounds for Proposition3.2.
>
> For Bayesian aggregation, we show that when $\displaystyle \min_{1\leq i\leq R}|p_i(x,a) - 1/2| > \epsilon$ :
>
> $$|\Delta_{DP}(\hat{Y}^{\phi}) - \Delta_{DP}(Y)| \leq 2(1-4\epsilon^2)^{\frac{R}{2}}$$
>
> This bound decay exponentially fast in $R$ for all finite crowd
> sizes, not just asymptotically.
>
> For Majority Vote the non-asymptotic bound is harder to establish (as for the asymptotic case), we are working on, changing the proof strategy.
>
> **L2:**
> The convergence follows from the convexity of the function $\mathcal{L}$ under sequential quadratic programming. The only aspect requiring clarification concerns the softmax approximation, which remains valid as long as the temperature constant $c$ is sufficiently small. More specifically, we have shown that the local minimum obtained via the softmax approximation incurs an error of order $O(c)$ relative to the ground‑truth optimum. In practice, the approximation gap is on the order of $10^{-4}$, which is negligible compared with the fairness tolerance threshold $\epsilon \geq 0.01$.
>
>
> **Time Complexity:**
>
> The overall algorithm complexity is
> $
> O\left((T + G^2) KN\right),
> $
> where: T denotes the number of SQP iterations, $G$ the grid  resolution, $N$ number of training samples and $K$ the number of classes
>
> Independently of the specific algorithm used for convex optimization (in our case, Sequential Quadratic Programming), the computational complexity depends on the number of constraints, as discussed in Denis et al., 2024, p.22.
>
> **L3:**
> In our experiments, we estimate the annotators’ confusion matrices under the assumption that
> $\pi^{(r)}_{y',y}(a)$ does not depend on the features.
> This limitation arises from the aggregation method’s modeling choice for the posterior estimates $\hat{\Phi}(w,a)$, rather than from our postprocessing method, which simply takes these posterior estimates as input and is therefore agnostic to such assumptions. Our framework does, in theory, allow feature‑dependent annotator skills (Theorem 3.1); in particular, $w$ may include $(\tilde{Y}, X)$ as discussed at the beginning of Section 4.
> Incorporating feature‑dependent annotator skill or bias would improve overall performance only if the posterior estimates $\hat{\Phi}(w,a)$ themselves become more accurate.
>
> In practice, however, reliably estimating feature‑conditional confusion matrices poses identifiability challenges when annotation redundancy is limited (e.g., the Jigsaw dataset has a median of only four labels per item), as also noted in the review of (Ibrahim et al. 2025). For this reason, we opt for a simpler approach in the experiments. Nonetheless, for completeness, we plan to include additional experiments in the appendix using recent crowdsourcing methods that incorporate both features and sensitive attributes.
>
> These points will improve the paper thanks to the reviewer comment.
>
> References:
> Denis, C., Elie, R., Hebiri, M., and Hu, F. Fairness guarantees in multi-class classification with demographic parity.
> Journal of Machine Learning Research, 25(130):1–46,
> 2024.
>
>
> Ibrahim, S., Traganitis, P. A., Fu, X., and Giannakis, G. B.
> Learning from crowdsourced noisy labels: A signal processing perspective. IEEE Signal Processing Magazine,
> 42(3):84–106, 2025

---

> > ### Author Rebuttal · Reviewer_wESF · 2026-04-04
> >
> > Thank you for the thorough rebuttal. I will maintain a positive attitude for this paper.

---

### Decision · Program_Chairs · 2026-04-30

**Decision:**

Accept (regular)

**Comment:**

The paper studies an important and relatively underexplored problem: fairness in crowdsourced label aggregation. Its strongest aspect is the theoretical contribution, including bounds on demographic parity gaps and an analysis of bias amplification. The proposed FairCrowd method is a flexible post-processing approach that can be applied across different aggregation rules, and the empirical results illustrate the expected fairness–accuracy trade-offs.

One limitation is the narrow focus on demographic parity, with limited discussion of extensions to other widely used fairness notions such as equalized odds or equal opportunity. In addition, the theoretical guarantees rely on simplified assumptions (e.g., annotator independence), and the paper does not sufficiently address their robustness in more realistic settings. That said, these concerns are relatively minor compared to the overall contribution.

Overall, the paper provides a meaningful theoretical and computational contribution to the study of fairness in crowdsourced aggregation, and I recommend acceptance.